

# Signs of climate variability in double tropopause global distribution from radio occultation data

Alejandro de la Torre[1], Peter Alexander[2], Torsten Schmidt[3], Andrea K. Steiner[4], Florian Ladstädter[4], Rodrigo Hierro[1],  Pablo Llamedo[5]

*Correspondence to*: Alejandro de la Torre (adelatorre@austral.edu.ar)

[1] LIDTUA (CIC) and CONICET, Facultad de Ingeniería, Universidad Austral, Mariano Acosta 1611, Pilar, Provincia de Buenos Aires B1629ODT,  Argentina

[2] IFIBA, CONICET, Ciudad Universitaria, C1428EGA, Buenos Aires, Argentina

10  [3] GFZ, GFZ German Research Centre for Geosciences, Section 1.1: GPS/Galileo Earth Observation, Telegrafenberg A17, D-14473 Potsdam, Germany

[4] Wegener Center for Climate and Global Change, University of Graz, Graz, Austria

[5] Departamento de Ciencias Aplicadas, Universidad Nacional del Alto Uruguay, Tejeda 1042, San Vicente N3364, Misiones, Argentina



**Abstract**

In a standard atmosphere, there is a single lapse rate tropopause (in what follows, tropopause) that separates the troposphere below from the stratosphere above. However, in certain situations, such as in regions of strong vertical wind shear or associated with certain weather phenomena, a second tropopause layer may form above the standard tropopause. The presence of a double tropopause (DT) can have implications for atmospheric and climate studies, as it may be associated with dynamic and complex weather patterns. Based on 14 years of temperature profiles retrieved by GNSS radio occultation and the resulting DT, a possible relationship between the spatio-temporal distribution of the relative number of DT to simple tropopauses (NDT) (*or dependent variable*) and a set of monthly climate indices (*or features*) is explored with a focus on the methodological approach. A cluster analysis is applied to geographically associate the DT occurrences with the climate indices. Then a multivariate linear regression is constructed using a progression of different models to identify the relevant features for the occurrence of DTs. On a global scale, from a hierarchical cluster analysis six sub-regions with different location and spread characteristics are identified. In addition to the condition of linearity in the residuals, the performance of each model in the train and test populations is evaluated to discard possible overfitting. The required conditions of non-collinearity, stationarity and cross-correlation between the features and the relative number of NDT after the removal of the climatological mean for each month (NDT') are checked. Mean squared errors, adjusted coefficient of determination (adjusted $R^2$) and number of degrees of freedom (F-statistic) parameters are evaluated for each model obtained. Taking into account the constraints of the present analysis, the most relevant climatic indices for the distribution of NDT' are identified.



## 1. Introduction

The atmosphere is conventionally divided into layers based on the vertical structure of the temperature
field. These layers, the troposphere, stratosphere, mesosphere and thermosphere, are separated by the
tropopause, stratopause and mesopause (e.g. Andrews et al. 1987). The tropopause is the boundary
between the troposphere, and the stratosphere, the layer above it. It is characterized by a temperature
inversion, where the temperature stops decreasing with altitude and remains relatively constant. A double
tropopause (DT) means the presence of two distinct tropopause layers in the Earth's atmosphere. In a
standard atmosphere, there is a single tropopause that separates the troposphere below from the
stratosphere above. However, in certain situations, such as in regions of strong vertical wind shear or
associated with certain weather phenomena, a second tropopause layer may form above the standard
tropopause. The presence of a double tropopause can have implications for atmospheric and climate
studies, as it may be associated with dynamic and complex weather patterns. At midlatitudes, the higher
tropical tropopause domain may overlap with the lower high-latitude tropopause domain and form double
tropopause (DT) occurrences, when the high-latitude tropopause domain extends equatorward or the
tropical tropopause domain extends poleward. DTs occurrence is associated with extratropical synoptic
disturbances, in storm track regions, on the poleward side of the subtropical jet stream frequently during
winter (Bischoff et al., 2007; Schmidt et al., 2006; Seidel and Randel, 2006). DTs have also been found in
association with mountain gravity waves (Schmidt et al., 2006), cyclogenesis (Añel et al., 2008), in strong
cyclonic circulation systems (Peevey et al., 2014; Wang and Polvani, 2011), linked to the strength of the
upward branch of the Brewer-Dobson circulation (Castanheira et al., 2012) and detected in cloud-top
inversion layers (Biondi et al., 2012). There is a general agreement that the existence and understanding
of DTs distribution provides important knowledge of the global distribution of stratosphere-troposphere
exchange.

Since more than two decades, globally distributed measurements from radio occultation (RO) are
available. This limb sounding technique provides temperature profiles with high accuracy and vertical
resolution in the troposphere and lower stratosphere for applications in atmospheric and climate research
(Anthes, 2011; Steiner et al., 2011). Wilhelmsen et al. (2020) presented global and seasonal
characteristics of DTs from RO observations. They analyzed the relation between DTs and El Niño
Southern Oscillation (ENSO) events and its implication on the tropopause structure based on a multiyear
RO record. The seasonal distribution of DTs revealed several hotspot locations, such as near the
subtropical jet stream and over high mountain ranges, where DTs occur particularly often. These authors
detected a higher number of DTs during the cold La Niña state while warmer El Niño events resulted in
lower DTs rates.

Machine learning (ML) enables computer systems to learn from information systems, making it possible
to forecast their variables and to detect patterns. This becomes even more relevant when considering that
ML can utilize information derived from big data analysis to produce better results. The development of
big data analysis has proven to be useful for studying various datasets from multiple sources. In terms of
forecasting of meteorological data, progressive validation is an approach for validating machine learning
models, wherein data is progressively incorporated into the model, and its performance is assessed at each



stage. This enables the understanding of how the model's performance evolves as additional data is integrated. Multivariate statistical analysis is widely used to analyze atmospheric variables like precipitation intensity and its spatial and temporal variability (Sarmadi and Shokoohi, 2015; Koroša and Mali, 2022). Cluster analysis enables to provide a physical classification of weather and climate patterns for several purposes (Strauss 2018), e.g., to provide a classification of the basic surface climates on earth (Netzel and Stepinski, 2016), to analyze typhoon tracks (Camargo et al. 2007) or to better understand and forecast the occurrence of weather extremes (storms, floods), climate extremes (heat waves, cold snaps) and other factors relating to human health (e.g., Cassou et al., 2005; Coleman and Rogers, 2007; Polo et al., 2011). In these applications, the aim is to reduce the large nominal dimensionality of the atmospheric dynamic and thermodynamic space to a manageable and discrete set of weather and climate states with predictive value.

In this study, starting from a database of DT obtained from RO observations, we propose to explore a possible relationship between the spatio-temporal distribution of DT and a set of monthly climate indices. The focus is on the methodological approach. We first apply a cluster analysis to geographically associate DT occurrences and then construct a multivariate linear regression using a progression of different models with train and test populations to identify climate indices relevant for DT occurrence. In Section 2, we describe the data and procedure to obtain those series as a function of latitude and longitude. Section 3 details the cluster analysis, both hierarchical and K-means, to define the geographical sub-regions. Section 4 summarises the restrictions, methodology and limitations to be considered in the multivariate linear regression models. Section 5 describes the applied model and section 6 presents and discusses the results obtained.

## 2. Double tropopause (DT) data

Global Navigation satellite system (GNSS) Radio Occultation (RO) is a well-established technique for obtaining global thermodynamic and dynamic information in the atmosphere. RO uses GNSS signals received by Low Earth-Orbiting (LEO) satellites for atmospheric limb sounding. Temperature ($T$) profiles are derived with high vertical resolution and global coverage under nearly any weather conditions, offering the possibility to carry out the global monitoring of the vertical $T$ structure, in particular the thermal tropopause. RO measurements from different missions are combined and used in continuation of each other (Foelsche et al., 2011). The Wegener Center OPSv5.6 data set (Angerer et al., 2017; EOPAC-Team, 2021) represents a compilation of most RO satellite missions from 2001 to 2020, enabling us to study DTs over a longer time period than previous analyses. In this study, we use temperature profiles from September 2006 to August 2020, interpolated to an evenly spaced, fixed vertical grid.

Following Wilhelmsen et al. (2020), we used the lapse rate tropopause algorithm to compute the first and second lapse rate tropopause for each RO temperature profile. We define the number of DT occurrences as $NDT \equiv N_2/N_1$, where N1 and N2 refer to the total number of single and double tropopauses detected in each cell. From September 2006 to August 2020 the time series of NDT, i.e., the dependent variable,



retrieved from radio occultation vertical profiles, were recorded. These were then pooled in 648 non
115 overlapping (10x10 degrees) cells between +/-180 and +/-90 longitude and latitude degrees respectively.

Our study was mainly motivated by exploring the relationship between the observed NDT time and space
variability and the variability of climate indices. We considered monthly time series of 29 global climate
indices (the independent variables *or features*, https://psl.noaa.gov/data/climateindices/list/) starting on
September 2006 (168 months).

120 Prior to carrying out this task, an inspection of NDT series at different geographic positions reveals a
complex pattern with a prevailing temporal variability that depends essentially on the latitude. This is
illustrated in Fig. 1, where a band of 18 cells centered at -175 deg longitude was chosen as an example:

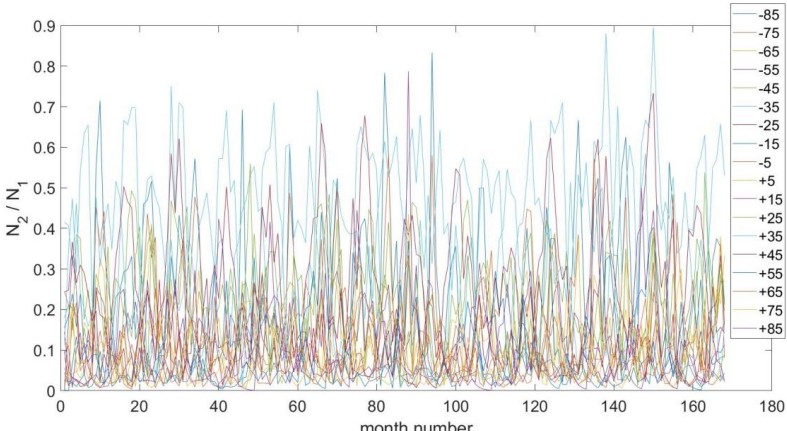

**Figure 1**. NDT meridional variability, arbitrarily chosen for the band of 18 cells at successive latitudes,
125 centered at -175 deg longitude.

Given this considerable variability and the objective of grouping time series with similar characteristics,
we developed a classification of NDT time series according to their cell latitude and longitude position,
applying cluster analysis (CA) techniques.

130 **3. Cluster analysis (CA)**

A CA assembles a set of objects (in this case time series) into groups (Andenberg 1973), where the
objects in the same cluster are similar and objects in different clusters are dissimilar (Balling, 1984). In
particular, the distance between objects is the similarity used to form the clusters. These distances
(similarities) are based on single or multiple K-dimensions. Each dimension represents a condition in
135 order to group the objects. In our case (see Fig. 1) an early inspection of the spread exhibited by i) mean
time values and ii) standard deviations over the 18 NDT time series suggest the choice of dimensions i)
and ii) and then K=2. In order to remove the seasonal variability, we computed the climatological



monthly means over 2006-2020 and subtracted it from each respective month in the NDT time series. The de-seasonalized time series is denoted as NDT'. The mean values of the NDT time series and the standard deviations of the NDT' time series are then used for the clustering. CA assigns group memberships at varying levels of aggregation and clusters should be comprised of points separated by small distances, relative to the distances between clusters. There are a wide variety of plausible definitions of distance in this context, and the results of a cluster analysis may depend quite strongly on the distance measure chosen. A given CA can be hierarchical and nonhierarchical (Wilks, 2006) depending on the way the clusters are formed. We briefly summarize both analyses, since we needed to apply them in a complementary way.

In a hierarchical agglomerative CA a hierarchy of sets of groups was constructed. Each group was formed by merging one pair from the collection of previously defined groups. This procedure begins by considering that the *n* individual observations have no group structure or, equivalently, that the data set consists of *n* groups containing one observation each. The first step is to find the two groups (i.e., data vectors) that are closest in their K-dimensional space and to combine them into a new group. In particular, the clustering in *n* groups at the beginning of this process and the one-group clustering at the end of it *are not* useful. A natural clustering of the data into a workable number of informative groups will emerge at some intermediate stage. There are alternative definitions for the above mentioned distances between groups of points if the groups contain more than a single member. We have chosen the so called average-linkage clustering. Each cluster-to-cluster distance is defined by the average distance between all possible pairs of points in two groups being compared. In our CA, we expect that the decided intermediate state should reveal the optimum number of clusters, each containing a number of cells that best represent the space and time patterns of DT time series.

The intermediate results of a hierarchical CA are illustrated using a dendrogram. An important practical problem in cluster analysis is the choice of which intermediate stage will be chosen as the final solution. That is, we need to choose the level of aggregation in the tree diagram at which to stop further merging of clusters. The principle guiding this choice is to find the level of clustering that maximizes similarity within clusters and at the same time, minimizes similarity between clusters. In Fig. 2, we note that selecting a cutoff distance = 0.07 suggests a reasonable preliminary number of six clusters in the dendrogram, while retaining a significant number of individual cells. We then proceed with the cluster analysis according to this classification in six groups.





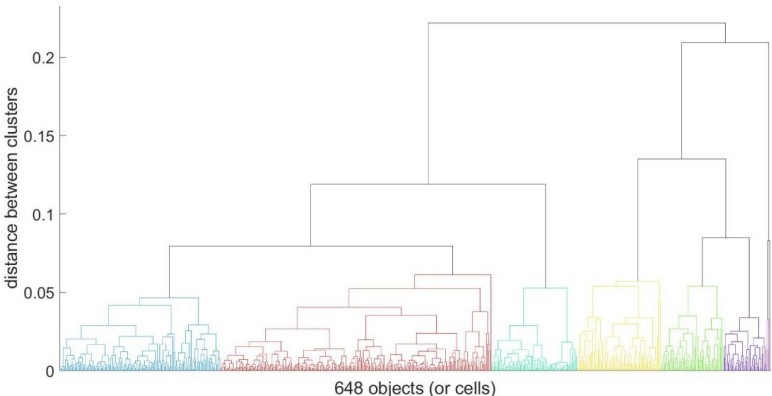

**Figure 2**. Dendrogram obtained by grouping 648 objects (or time series) into 6 clusters (to differentiate
each cluster from the others, different colors were arbitrarily chosen). Each time series corresponds to a
different geographical cell.

In a nonhierarchical CA we start from this previous preliminary observation. Using K-means, the 648
cells were pooled into 6 sub-regions. In Fig. 3, the cluster assignments and their respective centroids
corresponding to each cluster are shown, based on the means of NDT and standard deviations of NDT'. A
reasonable separation of objects is achieved.

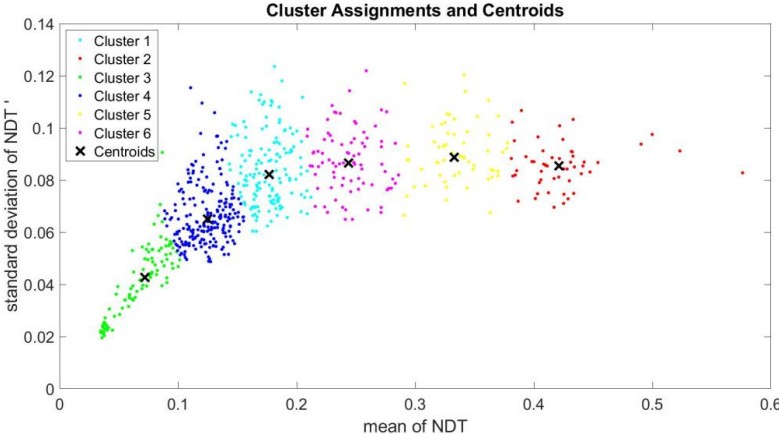

**Figure 3**. K-means clustering the 648 cells were pooled into 6 sub-regions. Here K = 2 dimensions are
the NDT mean value and NDT' standard deviation over the 168 months considered.



In Fig. 4, the time series belonging to each of the clusters, averaged over the 168 months are shown. A
clear distinction among the 6 curves evidences a convenient grouping in both K dimensions, according to
diverse geographic sectors.

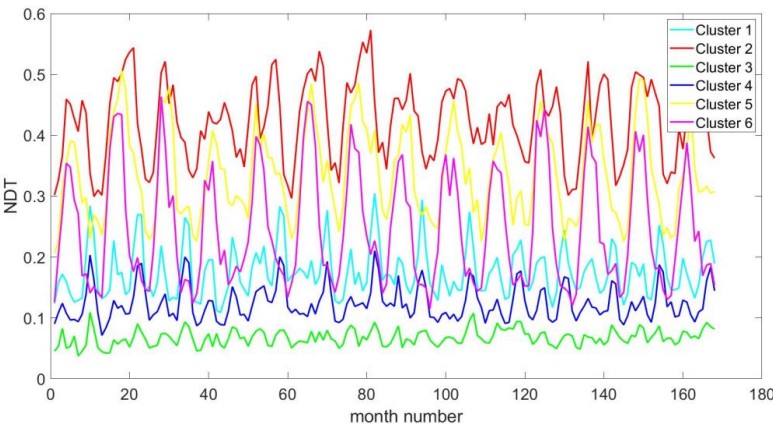

**Figure 4**. NDT averaged over all time series within each cluster.

An inspection of Fig. 4 evidences some characteristics which may be described into two groups: In
clusters 2, 5 and 6, NDT is dominated by a clear annual oscillation, with the highest mean values, the
highest dispersion and a less evident semiannual contribution. In clusters 1, 3 and 4, NDT exhibits a
considerably lower dispersion, lower mean values and an annual and semiannual variability in addition to
a complex superposition of several harmonics.

Given the variability of NDT for each group, we removed from NDT the climatological mean for each
month corresponding to the 14-year data set. The anomaly time series NDT' is shown in Fig. 5.

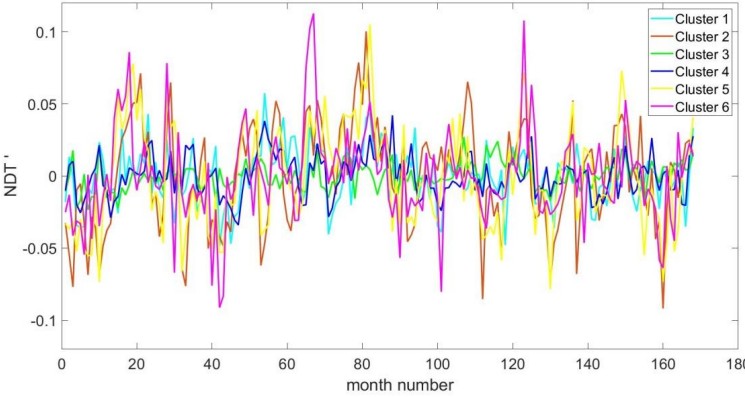

**Figure 5.** NDT' time series for each of the 6 clusters, after the removal of the NDT monthly mean
computed from 14-year data.



Here the amplitude variability of NDT' in sub-regions 2, 5, and 6 is clearly larger than in 1, 3 and 4,
consistently with Fig. 4 after the elimination of the annual contribution.

The geographic distribution of each cluster is shown in Fig. 6, where the colors used to distinguish each
of them were preserved.

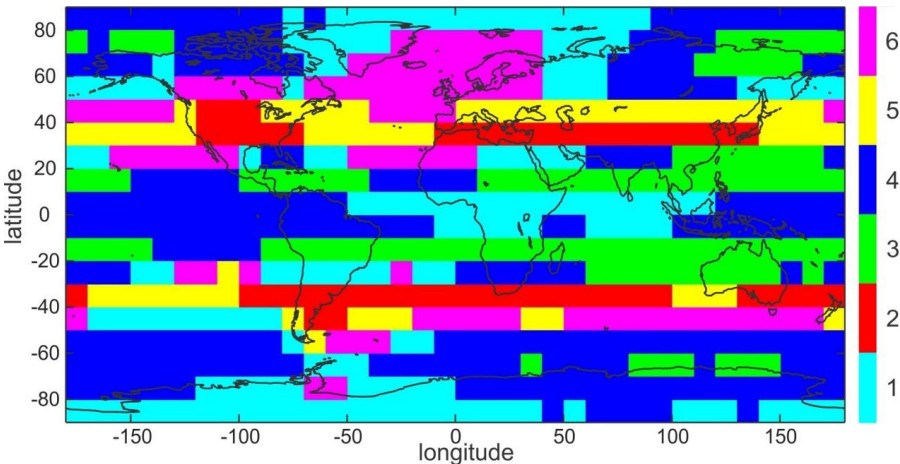

**Figure 6**. Cluster distribution obtained after the K-means analysis.

In Fig. 6, an attempt was made to optimize a balance between i) a significant number of cells belonging to
each sub-region and ii) their connexed structure (here we understand that in a connexed sub-region every
cell belonging to the sub-region should be ideally connected to the rest of its cells too). A prevailing
latitudinal stratification of the 6 clusters considerably symmetric with respect to the equator is evident.
The interconnected nature of each sub-region is highlighted in the polar, sub-polar and equatorial regions
by clusters 1 and 4. Clusters 3 and 2 are quite symmetrically distributed around the equator, at subtropical
and mid-latitudes respectively, while sub-region 5 is more dispersed between 30° and 50° also in both
hemispheres and it does not exhibit a homogeneous behavior. There is a prevailing clustering above
oceanic areas in subregion 6.

### 4. Conditions to multivariate regression models and methodology


*1. Non-collinearity*

We propose to analyze the possible existence of patterns relating the dependent variable (NDT') and
climate indices, by means of a multivariate linear regression.

To achieve this, we start by checking the condition of non-collinearity among the proposed features. In
doing so, the Pearson correlation matrix, including the climate indices provided in
https://psl.noaa.gov/data/climateindices/list/, was calculated. Highly correlated features are considered to
have values above 0.75 and are therefore interpreted as representing the same climate variability. Of the





climate indices that show a high correlation, only one is taken into account and retained. The climate
index finally selected is the one that shows the highest correlation with NDT'. Following this procedure,
the number of climate indices was reduced from 29 to 18 and they are listed in Appendix A. As an
example, from the climate indices describing El Niño variability, the Meiv2 index was retained.

  *2.  Stationarity*

A second condition that restricts the application to a multivariate linear regression is the stationarity of the
time series involved. Time series are considered as stationary when their statistical properties do not
change over time. The mean, variance, and autocorrelation structure must remain constant across different
time periods exhibiting repeating patterns at regular intervals. However, the presence of periodicity does
not necessarily indicate stationarity. In some cases, even though a time series has periodic patterns, it
might still exhibit trends. To determine whether the periodic time series are stationary, we perform a test
for stationarity to NDT' and each feature: the Augmented Dickey-Fuller (ADF) test (Dickey and Fuller,
1979). According to it, the rejection of the null hypothesis allows to discard a non-stationary character of
the series. In our case, by setting a significance level of 0.95, 12 indices succeeded to reject the null
hypothesis. As mentioned above, the climate indices or features were common to all 6 clusters. However,
from the same test we observed that those series for clusters 1 to 6 required a relaxation of the
significance level to 0.90 to reject the null hypothesis. Extending this less restrictive condition also to the
features, we concluded the suitability of including 3 additional time series, reaching a final number of 15
climate indices (highlighted in italic in Appendix A).

  *3.  Lagged features*

A third consideration is the following. Time series data often have mutual lags or time dependencies, so it
is worth considering possible time lags between predictors and response variables. Cross-correlations
(CC) between each of the features and the NDT' series in each of the sub-regions are calculated in order
to investigate for which lag (k) ranging from k = +5 to - 5 months these correlations are the highest.
These k values may be positive or negative. For a given k,  in general different for each feature, a relative
maximum positive or negative CC is evidenced between NDT' and each lagged feature. In Fig. 7, CC is
illustratively shown for sub-region 1 after the shifting process (see in Appendix B the corresponding CC
plots for the remaining sub-regions).



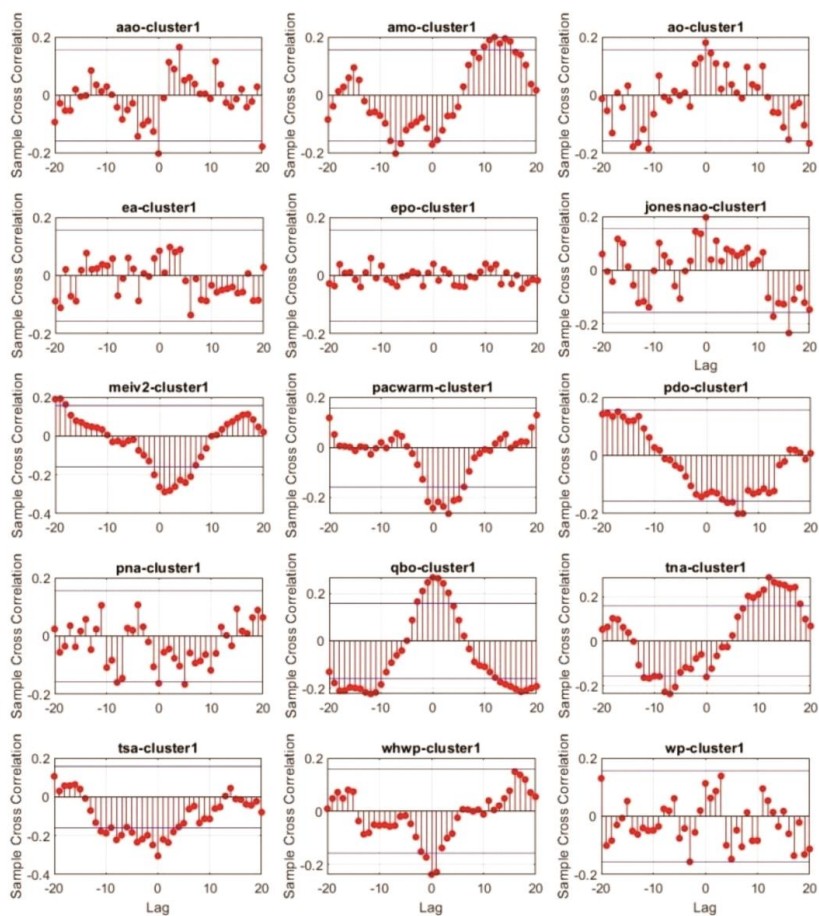

**Figure 7.** CC between the selected 15 features and DT in sub-region 1, after the shifting process (see
text).

After having shifted each feature, a relative maximum cross-correlation at zero lag is found. The time lags
detected on the basis of shifted CC in the 6 sub-regions are detailed in Table 1. We observed that in some
clusters some features were not shifted.

| Sub-region | Aao | Amo | Ao | Ea | Epo | Jonesnao | Meiv2 | Pacwarm | Pdo | Pna | QBO | Tna | Tsa | Whwp | Wp |
|---|---|---|---|---|---|---|---|---|---|---|---|---|---|---|---|
| 1 | -4 | -2 | +2 | 0 | 0 | +2 | -5 | 0 | -5 | +1 | -4 | -2 | +2 | -3 | +2 |
| 2 | +4 | +1 | -4 | -1 | 0 | -1 | +5 | +1 | +1 | +2 | 0 | +1 | +4 | +1 | +2 |
| 3 | -5 | -1 | -1 | 0 | 0 | -1 | 0 | +4 | +2 | +4 | 0 | 0 | +4 | +1 | 0 |
| 4 | +4 | +2 | -1 | 0 | 0 | +1 | 5 | 0 | 0 | +4 | +5 | -4 | -2 | +3 | +1 |
| 5 | +5 | -4 | 0 | -2 | 0 | -2 | +5 | 0 | +2 | +2 | 0 | +2 | 0 | +1 | 0 |
| 6 | +3 | -4 | +1 | +1 | 0 | +1 | +5 | 0 | 0 | +2 | 0 | +2 | +4 | +1 | 0 |



**Table 1.** The time lags considered on the basis of the shifted CCs between the features and NDT' in the 6 sub-regions (see text).

From Table 1, we see, e.g., that QBO maximizes its CC with NDT' for lag=+5 in sub-region 4. This means that the Quasi-biennial Oscillation (QBO) is 5 months ahead of NDT' in this sub-region.

We propose below a linear model for each cluster considering each feature shifted in time backward or forward up to 5 months, consistently with the lag that maximizes the positive or negative CC within +5 and -5. We interpret a lag greater or less than 0 in a sub-region as maximising the CC between the respective feature and NTD', k months shifted earlier or later respectively.

In Fig. 8, the Pearson correlation matrix for sub-region 1 with NDT' and the selected 15 climate indices
detailed in Appendix A is illustratively shown to provide a first inspection of a possible relationship between NDT' and each feature. As mentioned before, the remaining rejected climate indices are strongly correlated with at least one of the features already included in Table 1 and Fig. 8. As can be seen in Fig. 8, there is a non-high correlation between any pair of features (it always remains below/above +/- 0.75). Similar values are obtained from a Spearman correlation matrix (not shown). In this sub-region, no single
correlation between any feature and NDT' is better than -0.30. As it will be shown below, a progressive addition of climatic indices to the model improves its performance.

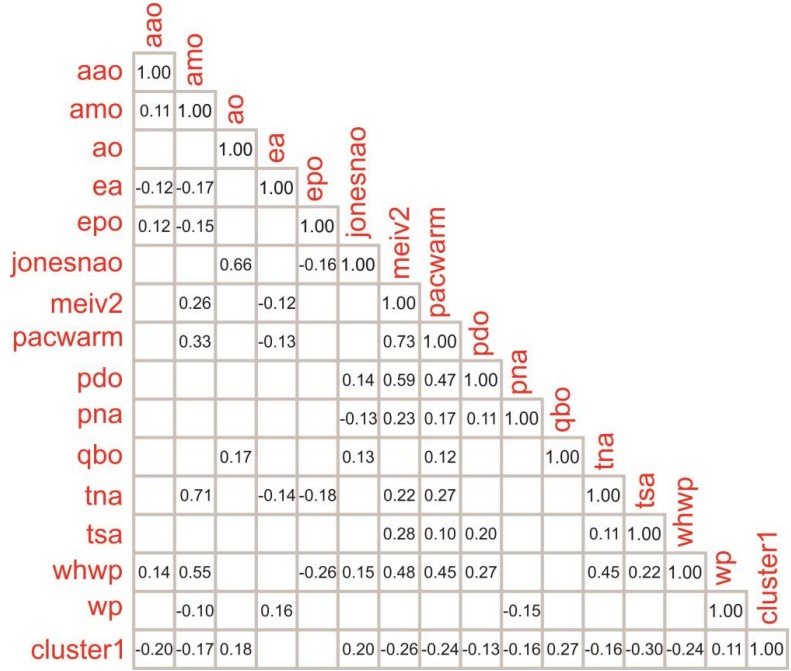

**Figure 8**. The Pearson correlation matrix for sub-region 1, retaining 15 climate indices and NDT' (=
"cluster 1").



*4.  Linear hypothesis*

A final restriction, whose accomplishment is to be later verified, requires that in any multivariate linear regression the residuals (the differences between the observed values and the predicted values) must follow a multivariate normal distribution. The assumption of normally distributed residuals is important

for making valid statistical inferences and conducting hypothesis tests. Normally distributed residuals are indicative of a well-fitting model. Deviations from normality may suggest that the model is not capturing all of the underlying structure in the data (Wilks et al 2006).

**5.  Building up the model**

We first split each data set at random into two distinct sets: training and testing (70% -30%).  The basic

idea behind this data grouping strategy is, that we learn the model parameters on the training set, and then we use the test data set (a data set that was not involved at all during parameter estimation) to determine the goodness of fit of our model and evaluate the generalization properties of the model (Hartmann et al 2023). It is expected that a progressive addition of climatic indices to the model improves its performance, reducing the root mean squared error (RMSE) obtained in training and testing sets, defined

as:

$$RMSE = \sqrt{\frac{\sum_{i=1}^{n}(pred_i - obs_i)^2}{n}} \qquad (1)$$

where $pred_i$ and $obs_i$ refer to each of $n$ observations.

We write the model as:

$$y_i = \beta_0 + \sum_{j=1}^{d} x_{ij}\beta_j + \varepsilon_i$$

$$= \beta_0 + x_{1i}\beta_1 + x_{2i}\beta_2 + \dots + x_{di}\beta_d + \varepsilon_i, \qquad (2)$$

$i$ =1, 2,….$n$ and $d$ is the number of features considered. $\varepsilon_i$ is the error of each observation. The $\beta$ coefficients are obtained by minimizing the residual sum of squares, thus obtaining:

$$\hat{\beta} = (X^T X)^{-1} X^T y \qquad (3)$$

Given new data $X_{new}$, the least squares prediction is:

$$\hat{y} = X_{new}\hat{\beta} = X_{new}(X^T X)^{-1} X^T y. \qquad (4)$$

We must use just as many features as necessary to achieve the best out-of-sample performance. The strategy to find the most useful feature sets constitutes our selection of features (Hartmann et al 2023). The price we have to pay by introducing feature selection approaches is that we loose the availability to judge the importance of particular features. Then, to find which features should be included in the model

we look for a criterion that allows to assess which combination of features gives the best model performance and, in order to counteract overfitting issues, penalizes the number of free parameters in our model (Hartmann et al 2023).



In doing so, we apply a model selection criterion, the Akaike Information Criterion (AIC). According to it, the relative quality of statistical models for a given set of data is measured. Given a collection of
models built up with different number of features, AIC estimates the quality of each model, relative to each of the other models. It provides a means for model selection. Under the framework of step-wise model selection a criterion, AIC is used for weighing the choices of adding (or excluding) model features, taking into account the numbers of features to be fitted. At each step, an add or drop is performed, that minimizes the criterion (AIC) score.

A forward-stepwise selection starts with the intercept (baseline model), and then sequentially adds into the model the predictor that most improves the fit. Forward-stepwise selection is a greedy algorithm, producing a nested sequence of models (Hartmann et al 2023). A backward-stepwise model selection is very similar to the forward-stepwise model selection. It starts with the full model, and sequentially deletes the predictor that has the least impact on the fit.

The evaluation of each one of the models' performance is made based on the following:

1) RMSE for training and testing sets.

If any of the models performs well on the training data set, but performs poorly on the test data set ($RMSE_{test} \gg RMSE_{train}$), this is an indication for model overfitting (an out of proportion between the number of features and observations). This is because we do not want our model to memorize the training
data set, but we want to find a model that accounts for the unknown data generation process generating our observations.

2) $R^2$, the coefficient of determination. It is the proportion of variance in the observed values explained by the regression equation. It is a statistical measure of how well the regression hyperplane approximates the real data points: a measure of the goodness of fit of the model: $R^2 = {SSR}/{SST}$, where $SSR =$
$\sum(\hat{y}_i - \bar{y})^2$ is the regression sum of squares representing the "explained" variance by the model, and $SST = \sum(y_i - \bar{y})^2$ is the total sum of squares or *total variance*. $\hat{y}_i$ and $y_i$ are, respectively, the modeled and measured NDT' values for each month. $0 < R^2 < 1$. A value of near 0 suggests that the regression equation is not capable of explaining the data. An $R^2$ of 1 indicates that the regression model perfectly fits the data. The addition of features to the model inflates the value of $R^2$. The addition of *d* features to a
regression model with *n* observations will increase the value of $R^2$, no matter how worthless the feature is. This issue is addressed by penalizing $R^2$ when parameters are added to the model. The result is an adjusted $R^2$, defined as:

$$adjusted\ R^2 = \left(R^2 - \frac{d}{n-1}\right)\left(\frac{n-1}{n-d-1}\right) \qquad (5)$$

When $R^2$ and adjusted $R^2$ are close to each another, this implies that the penalty for an increased number
of features is not very high. In our case, *n* is the number of months.



3) The F-statistic is the number of degrees of freedom. When the F-statistic is large, the *explained variance* (*SSR*) is large relative to the *unexplained variance* (*SSE*). It is an indicator that the selected features are useful variables to explain the response variable. $SSE = \sum(y_i - \hat{y}_i)^2$.

**6. Model results and discussion**

We proceed with two steps: i) we build the best NDT' model for each cluster and, from the best model found, ii) we detect which features are particularly relevant in each sub-region. Based on the methodology described in the previous section and the AIC criteria, in each sub-region we build a hierarchy of models applied to training and testing data to select the more appropriate features in each
cluster.

We then repeat the following process for the 6 sub-regions. After splitting the data set for each cluster into training and testing sub-sets we consider the *baseline* model as a reference. This is the simplest model we may construct in terms of RMSE. Any additional model from Eq. (2) should always perform better than the baseline model. In our example the baseline model is just the arithmetic mean of the
response variable. This arithmetic mean produces the same number as $\beta_0$ in Eq. (2).

To construct and compare with the rest of the models, the AIC model selection criterion is used for the case of forward-stepwise selection, which starts from the baseline model ($\beta_0$). The set of features in Eq. (2) that minimizes RMSE is detected, considering all possible linear models and combinations. This model is called "*with ALL*" (Fig. 9 and Table 2). Once the optimal set of $h_i$ (i=1 to 6) features (different
for each cluster) has been found, the relative influence of each of the climate indices in each sub-region is identified. For every cluster, additional $h_i$ models are constructed, but each containing $h_i$-1 features. These are respectively referred to as "*without* [*feature removed*]", given the disregard of each of $h_i$ features alternatively. Both the model *with ALL* and the models built from the removal of each of the features are produced from the training and test data sets to detect possible overfitting effects (Fig. 9).






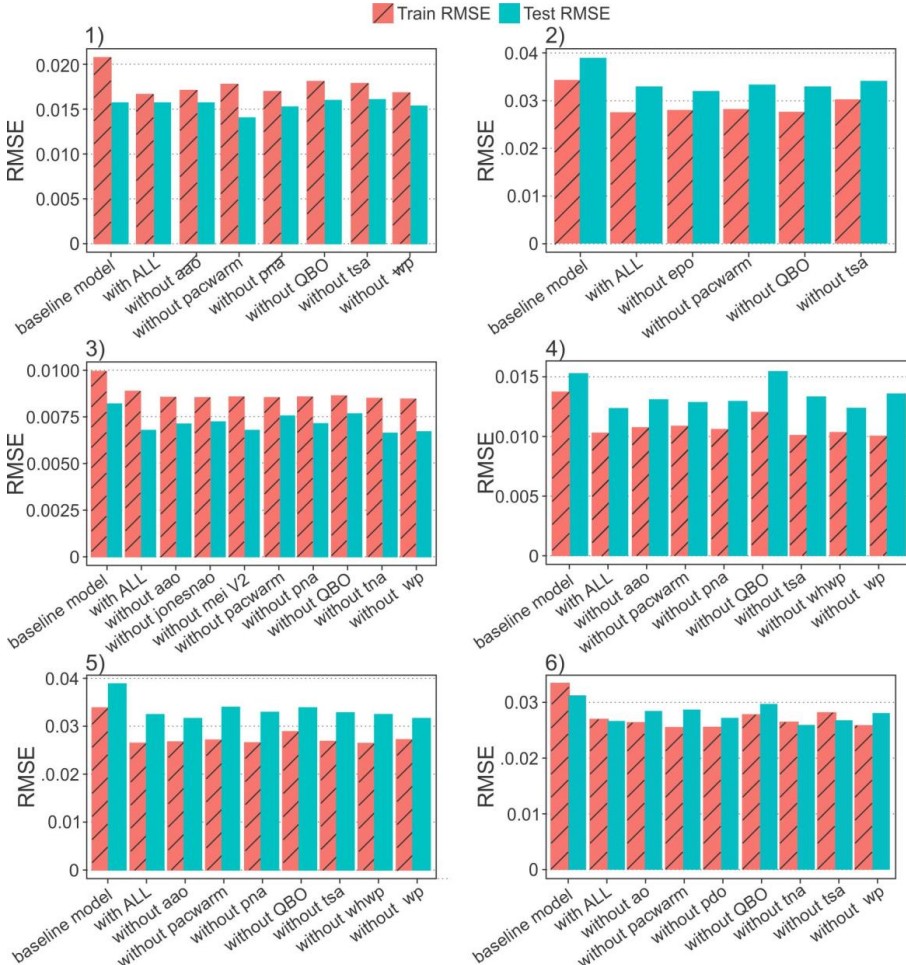

**Figure 9.** Baseline model, model *with ALL* features, and remaining models after disregarding each of features contained in *with ALL* model alternatively. 1) to 6) refer to clusters 1 to 6 respectively.

Figure 9 shows, as expected, the highest RMSE values corresponding to the *baseline* model. Different features were selected in each of the models by the AIC criteria and contrasted with the *baseline* and *with ALL* models for each cluster. Those features selected by the AIC criteria in each cluster are assumed to have a relationship with NDT' in that sub-region. The coefficients in Eq. (2) corresponding to the forward step-wise model in clusters 1 to 6 are shown in Appendix C. RMSE is shown for the training and test

samples. As explained before, this is an indication of the degree of possible model overfitting ($RMSE_{test} \gg RMSE_{train}$). Although the predictive capacity of the model in the context of a different NDT' data series is not a priority objective of this work, from an inspection of Fig. 9 we can point out that a possible indication of overfitting appears in clusters 2, 4 and 5.



It is important to underline that the present analysis is based on different assumptions, one of which is the proposed number of clusters. This arises from the hierarchical analysis, which was performed taking into account the available time span of the data series. Any increase in the number of clusters and the length of the time series would reduce the geographical extent of each sub-region, perhaps allowing the influence of each climate index on each cluster to be better specified. Moreover, from Fig. 8 and the corresponding ones for the remaining clusters (not shown), although some features individually have a non-negligible

correlation with NDT' they are not necessarily selected to build the best model *with ALL*.

Bearing in mind the constraints of the present analysis, some considerations can be made:

By inspection of Fig. 9, we observe the enhanced relevance of QBO to the variability of NDT' in clusters 4, 5 and 6, based on the relative increase in RMSE when this feature is removed from the model. Similarly, QBO appears, in both linear correlation with NDT' (Pearson index=0.48 in cluster 4) and AIC

analyses, as a significant feature mainly in tropical latitudes. An influence of QBO on the extratropical region has been referred to as the Holton–Tan effect, manifested as a weaker and warmer winter Arctic polar vortex during the easterly phase of the QBO (Holton and Tan 1980; Silverman et al 2021). According to AIC, Meiv2 influences the variability of NDT' at subtropical and middle latitudes in both hemispheres (cluster 3 in Fig. 9). Also a correlation between them is found in clusters 5 and 6 (Pearson

index=-0.38 and -0.33 respectively). A clear signal of QBO and ENSO from reanalyses and radio occultation data was previously reported in NDT on a global scale (Castanheira et al. 2012; Wilhelmsen et al. 2020). Some of the DTs detected in the tropics are related to changing QBO phases (e.g., Kedzierski et al., 2016). A relationship between DTs and ENSO was detected along -40°S and 40°N as well as north of 40°N between 60°E to 60°W (Wilhelmsen et al., 2020; Fig. 3), which is in agreement with our clusters

5 and 6. Pacwarm is included by AIC in all clusters, however, it has the highest Pearson correlation with NDT' in sub-regions 5 and 6. From AIC, the Arctic oscillation (Ao) is relevant in region 6 while the tropical southern Atlantic index (Tsa) appears relevant in regions 2. These features, together with those selected by the AIC analysis in Fig. 9 are considered to be linearly related to the distribution of NDT'.

As for the remaining features not mentioned in Section 4 and not reported in the AIC analysis, we

consider them not relevant in relation to the variability of NTD'. To summarize, Table 2 shows, in addition to those features included according to the AIC forward step-wise method, $R^2$, adjusted $R^2$ and F-statistic for *with ALL* models (AIC selection) and each cluster.






| Sub-region | Aao | Amo | Ao | Ea | Epo | Jones-nao | Meiv2 | Pac-warm | Pdo | Pna | Qbo | Tna | Tsa | Whwp | Wp | $R^2$ | Adj- $R^2$ | F-st |
|---|---|---|---|---|---|---|---|---|---|---|---|---|---|---|---|---|---|---|
| 1 | X | | | | | | | X | | X | X | | X | | X | 0.36 | 0.32 | 9.75 |
| 2 | | | | | | | | X | | | X | | X | | | 0.35 | 0.33 | 14.37 |
| 3 | X | | | | | X | X | X | | X | X | X | | | X | 0.29 | 0.24 | 5.24 |
| 4 | X | | | | | | | X | | X | X | | X | X | X | 0.49 | 0.45 | 13.86 |
| 5 | X | | | | | | | X | | X | X | | X | X | X | 0.42 | 0.38 | 9.25 |
| 6 | | | X | | | | | X | X | | X | X | X | | X | 0.43 | 0.39 | 11.12 |

**Table 2.** Features included in "*with ALL*" models in clusters 1 to 6 and the resulting $R^2$, adjusted $R^2$ and F-statistic parameters.

It may be observed that in general, the relative difference between $R^2$, adjusted $R^2$ reveals a low penalty
for an increased number of features. The highest values of the adjusted $R^2$ are in clusters 4, 5 and 6. The
F-statistic, in general is high. Accordingly, the explained variance is large relative to the unexplained
variance and the selected features are useful variables to explain NDT'. Finally we observe that the
RMSE of the test sample in clusters 1 and 3 in Fig. 9 is lower than that of the training sample. To explain
this, if the test data are similar to the training data but without as much noise, the model may perform
better in the test population, resulting in a lower RMSE. Moreover, if the training data contain many
outliers that negatively affect the model, and those outliers are not present in the test data, the RMSE may
be higher in the training data than in the test data.

Figure 10 shows the residual distributions obtained from *with ALL* models. A fair Gaussian distribution
supporting the linear hypothesis of these models is observed in clusters 1, 2 and 3 and some bias is
evident in sub-regions 4, 5 and 6.





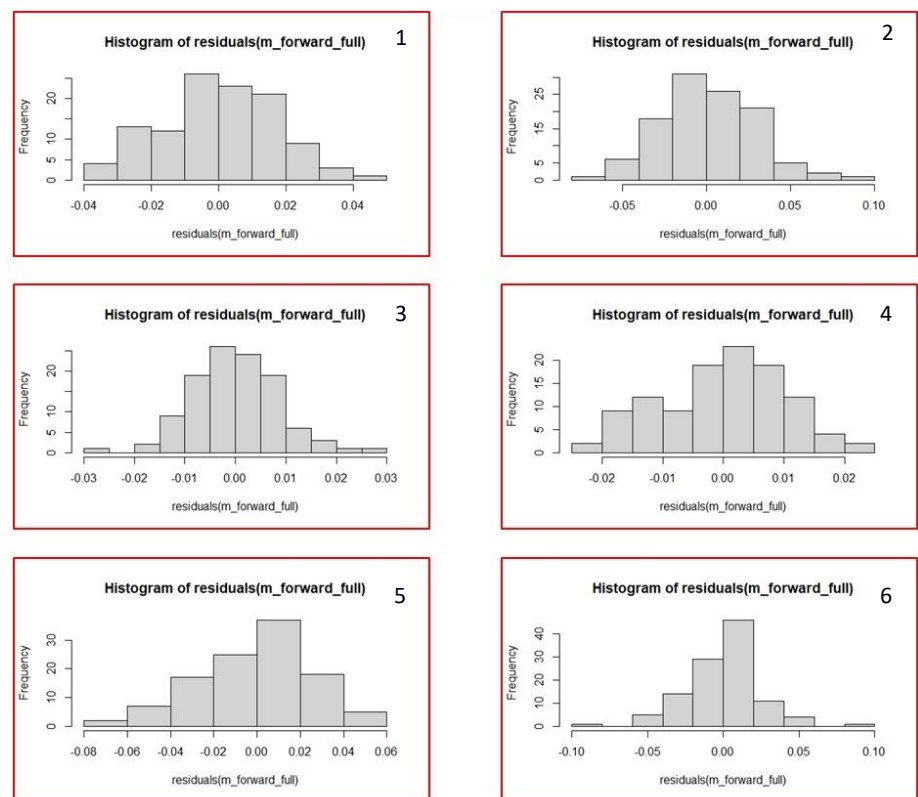

**Figure 10.** Residual distributions obtained from the *with ALL* models. 1) to 6) refer to clusters 1 to 6 respectively.

Finally, in Fig. 11 the corresponding Q–Q (quantile–quantile) plot to each cluster to reject possible
departures from the theoretical normal distribution is shown. These departures appear as evident only in clusters 5 and 6. As it is known, a Q–Q plot is a scatterplot where each coordinate pair consists of a data value and the corresponding estimate for that data value derived from the quantile function of the fitted (here Gaussian) distribution. It may be used to compare the shapes of distributions, providing a graphical view of how properties such as location, scale and skewness are similar or different in both distributions
(Wilks et al., 2006). It is worth mentioning that the application of the *backward* rather than *forward* step-wise model to the clusters does not reveal significant differences to Fig. 9 (not shown).



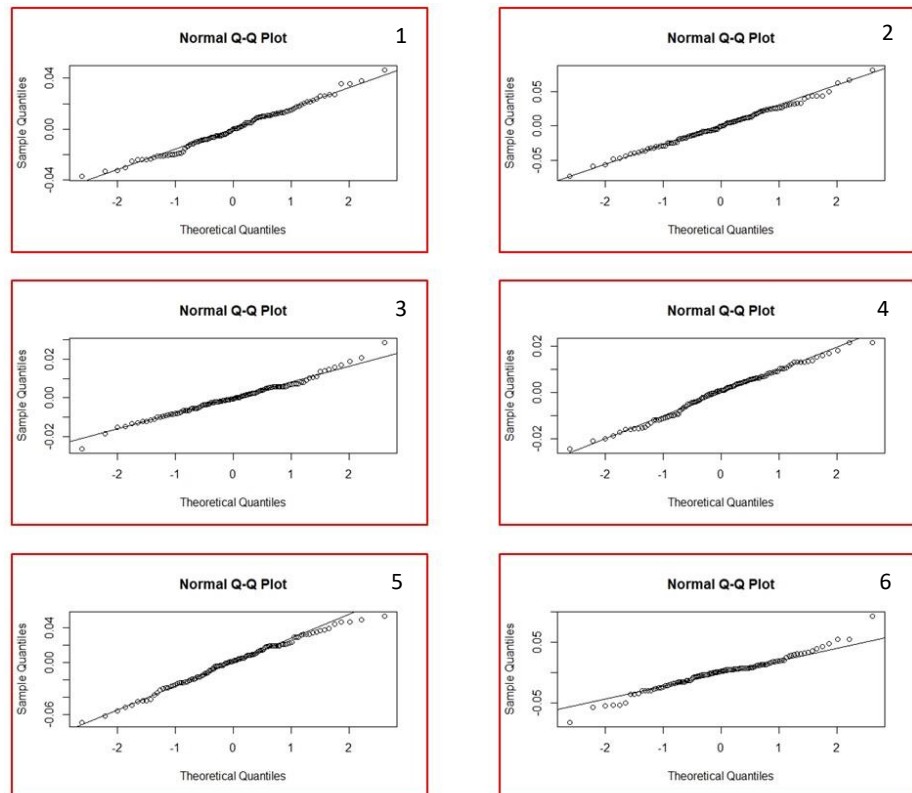

**Figure 11.** Q-Q plots in sub-regions 1 to 6 to detect departures from the theoretical normal distribution.

In this analysis, the resulting NDT' clustering was defined on the basis of two measures of location and spread in the time series: mean and standard deviation. Higher order moment that would account for the symmetry of NDT' could also have been included too. Additional variations may be considered, such as non-linear models, different values for the significance thresholds of the stationarity condition, maximum lags, a different proportion of data between the training and test populations, and possible relaxation of a strictly linear model. In this case, polynomial, hybrid, ridge or Lasso regressions could be addressed in the analysis of groups 5 and 6.

**Author contribution**

AT, PA and RH: Conceptualization, methodology, software. TS, AS and FL: Analysis, resources, data curation and supervision. AT prepared the manuscript with contributions from all co-authors. PL: review and editing.

**Competing interests**

The authors declare that they have no conflict of interest.



**Acknowledgements**

Manuscript prepared under grants CONICET PIP 11220210100292CO and ANPCYT PICT-
434 2021-I-A-01259. A.T., P.A., R.H. and P.L. are members of CONICET. A.K.S. and F.L. acknowledge
the financial support by the University of Graz.

**Data availability statement**

WEGC GNSS RO OPSv5.6 data are available online at https://doi.org/10.25364/WEGC/OPS5.6:2021.1/.

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



**Appendix A:** Climate indices (those highlighted in italic were detected as stationary).

1. ***Aao:** (Antarctic oscillation) is a low-frequency mode of atmospheric variability of the southern hemisphere that is defined as a belt of strong westerly winds or low pressure surrounding Antarctica which moves north or south as its mode of variability*

2. ***Amo:** (Atlantic Multidecadal Oscillation) (obtained from http://www.esrl.noaa.gov/psd/data/correlation/amon.**us**.long.data). The time series are calculated from the Kaplan SST dataset which is updated monthly. It is basically an index of the N Atlantic temperatures. Time series are created; a smoothed version and an unsmoothed version. In addition, two files starting at 1948 are produced to be used in the Correlation webpages.*

3. ***Ao:** (Artic Oscillation) The daily AO index is constructed by projecting the daily (00Z) 1000mb height anomalies poleward of 20°N onto the loading pattern of the AO. The loading pattern of the AO is defined as the leading mode of Empirical Orthogonal Function (EOF) analysis of monthly mean 1000mb height during 1979-2000 period.*

4. ***Ea**: The positive phase is associated with positive height anomalies located over Europe and northern China, and negative height anomalies located over the central North Atlantic and north of the Caspian Sea.*

5. ***Epo** (or **Ep**): The East Pacific - North Pacific (EP- NP) pattern is a Spring-Summer-Fall pattern with three main anomaly centers. The positive phase of this pattern features positive height anomalies located over Alaska/ Western Canada, and negative anomalies over the central North Pacific and eastern North America.*

6. **Gmmsst.** Gaussian Global Model Sea Surface Temperature.

7. ***Jones NAO.** North Atlantic Oscillation (by Jones). The Jones index is the difference in atmospheric pressure between Gibraltar, situated within the zone of impact of the Azores' anticyclone, and south-western Iceland (Stykkisholmur/Reykjavik) (Jones et al. 1997). Jones P. D., Jonsson T., Wheeler D., 1997, Extension to the North Atlantic Oscillation using early instrumental pressure observations from Gibraltar and South-West Iceland. Int J. Climatol. 17: 1433-1450.*

8. ***Meiv2.** Leading combined Empirical Orthogonal Function (EOF) of five different variables (sea level pressure (SLP), sea surface temperature (SST), zonal and meridional components of the surface wind, and outgoing longwave radiation (OLR)) over the tropical Pacific basin (30°S-30°N and 100°E-70°W).*

9. ***Pacwarm.** 1st EOF timeseries of SST (60°E-170°E, 15°S-15°N) SST EOF, all months*

10. ***Pdo:** (Pacific Decadal Oscillation). PDO is the leading PC of monthly SST anomalies in the North Pacific Ocean.*

11. ***Pna.** is one of the most prominent modes of low-frequency variability in the Northern Hemisphere extratropics.*

12. ***Qbo**. Zonally averaged equatorial wind data at 50 hPa from the NCEP/NCAR R-1 reanalysis.*

13. **Solar.** Solar Flux (10.7cm).

14. ***Tna:** Tropical Northern Atlantic Index. Anomaly of the average of the monthly SST from 5.5N to 23.5N and 15W to 57.5W. HadISST and NOAA OI 1x1 datasets are used to create index.*

15. ***Tni.** Trans Niño index.*

16. ***Tsa:** Tropical Southern Atlantic Index. Anomaly of the average of the monthly SST from Eq-20S and 10E-30W. HadISST and NOAA OI 1x1 datasets are used to create index*

17. ***Whwp:** Western Hemisphere Warm Pool. Monthly anomaly of the ocean surface area warmer than 28.5° C in the Atlantic and eastern North Pacific.*

18. ***Wp:** Western Pacific Index. The WP pattern is a primary mode of low-frequency variability over the North Pacific in all months.*













**Appendix B: CC** between the selected 15 features and DT in sub-regions 2 to 6 (Figures A1 to A5 respectively), after the shifting process.

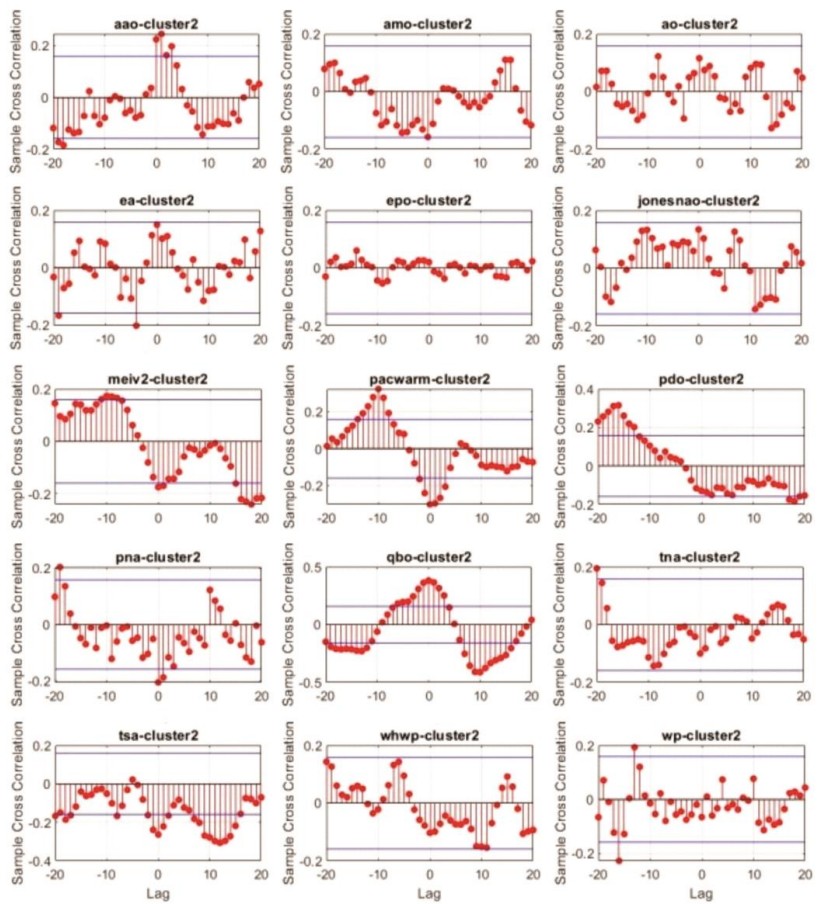

**Figure A1.** CC between the selected 15 features and DT in sub-region 2, after the shifting process (see text).



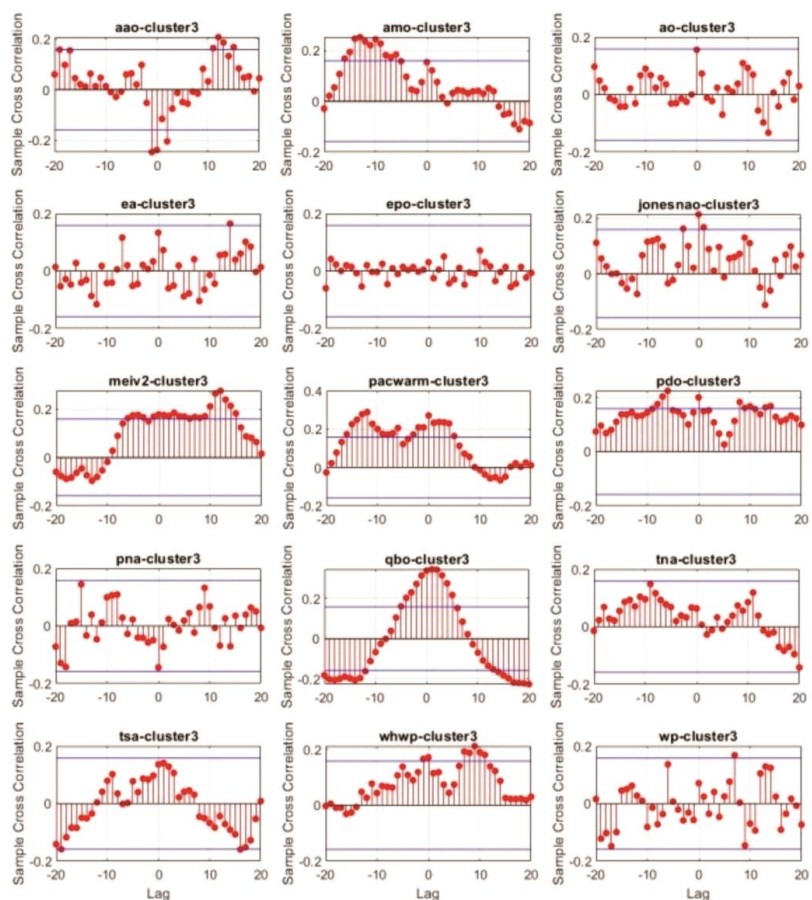

**Figure A2.** Same as in Fig. A1, in sub-region 3.






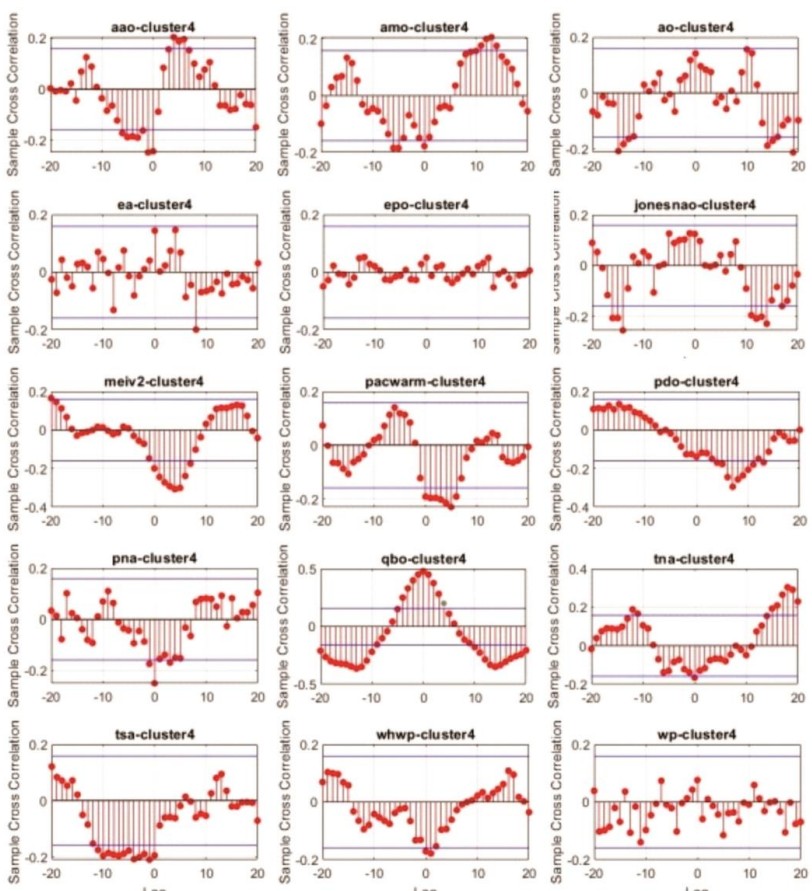

**Figure A3.** Same as in Fig. A1, in sub-region 4.




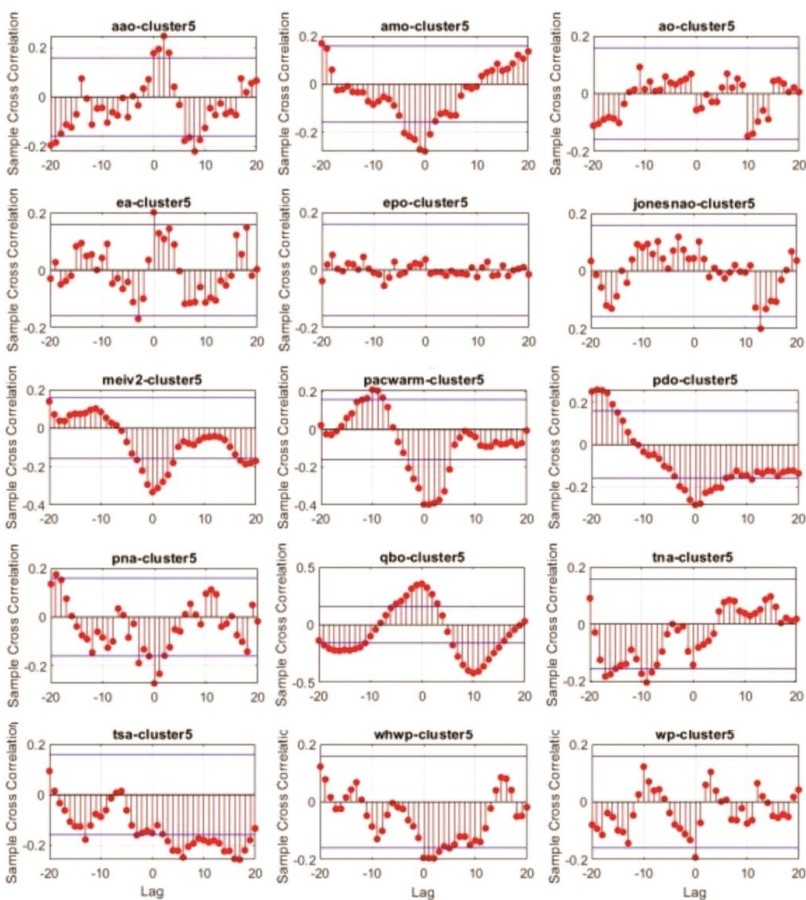

**Figure A4.** Same as in Fig. A1, in sub-region 5.






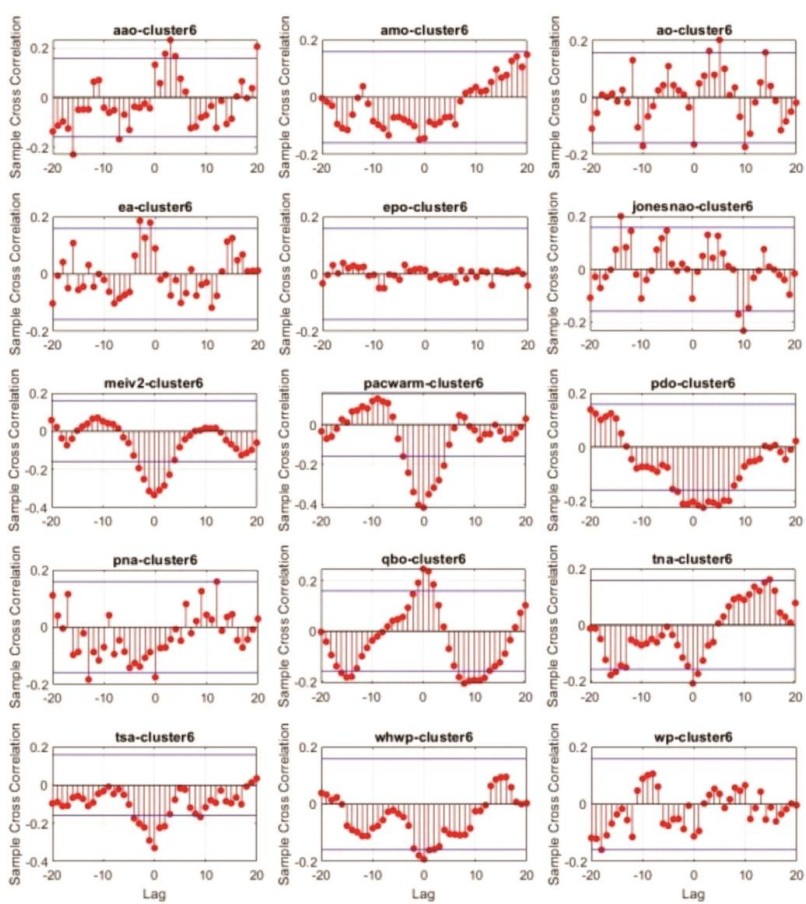

**Figure A5.** Same as in Fig. A1, in sub-region 6.









**Appendix C:** Coefficients and features corresponding to the "*with ALL*" model in clusters 1 to 6.

```
    cluster1 ~ tsa + qbo + pacwarm + aao + pna + wp
```

```
Residuals:
      Min        1Q    Median        3Q       Max
-0.036895 -0.010484  0.000123  0.011318  0.046354
```

```
Coefficients:
```
```
              Estimate Std. Error t value Pr(>|t|)
(Intercept)  0.0194782  0.0034257    5.686 1.18e-07 ***
tsa         -0.0208061  0.0051643   -4.029 0.000106 ***
qbo          0.0004799  0.0001098    4.369 2.94e-05 ***
pacwarm     -0.0341242  0.0090960   -3.752 0.000288 ***
```
```
aao         -0.0039275  0.0016474   -2.384 0.018918 *
pna         -0.0037843  0.0017543   -2.157 0.033278 *
wp           0.0022488  0.0015611    1.440 0.152704
---
Signif. codes:  0 '***' 0.001 '**' 0.01 '*' 0.05 '.' 0.1 ' ' 1
```
```
Residual standard error: 0.01714 on 105 degrees of freedom
Multiple R-squared:  0.3579, Adjusted R-squared:  0.3212
F-statistic: 9.753 on 6 and 105 DF,  p-value: 1.587e-08
```

```
    cluster2 ~ qbo + tsa + pacwarm + epo
```

```
Residuals:
      Min        1Q    Median        3Q       Max
```
```
-0.072798 -0.017904 -0.000591  0.020900  0.080895
```

```
Coefficients:
              Estimate Std. Error t value Pr(>|t|)
(Intercept)  0.0335452  0.0055860    6.005 2.72e-08 ***
```
```
qbo          0.0012358  0.0001995    6.196 1.13e-08 ***
tsa         -0.0493305  0.0105276   -4.686 8.32e-06 ***
pacwarm     -0.0326702  0.0153436   -2.129   0.0355 *
epo          0.0001765  0.0001111    1.589   0.1151
---
```
```
Signif. codes:  0 '***' 0.001 '**' 0.01 '*' 0.05 '.' 0.1 ' ' 1
```

```
Residual standard error: 0.02817 on 106 degrees of freedom
Multiple R-squared:  0.3516, Adjusted R-squared:  0.3271
F-statistic: 14.37 on 4 and 106 DF,  p-value: 2.097e-09
```
```
    cluster3 ~ qbo + pacwarm + meiv2 + aao + jonesnao +
       pna + tna + wp
```

```
Residuals:
       Min         1Q     Median         3Q        Max
-0.0262363 -0.0053534 -0.0004189  0.0055748  0.0285033
```

```
Coefficients:
```
```
               Estimate Std. Error t value Pr(>|t|)
(Intercept) -2.375e-03  1.729e-03   -1.373   0.1726
qbo          1.426e-04  6.104e-05    2.336   0.0214 *
pacwarm      9.171e-03  5.124e-03    1.790   0.0765 .
meiv2        2.054e-03  9.667e-04    2.124   0.0361 *
```
```
aao         -2.120e-03  9.762e-04   -2.172   0.0322 *
jonesnao     9.833e-04  4.462e-04    2.204   0.0298 *
```





```
     pna          -1.679e-03  8.620e-04  -1.948    0.0542 .
     tna           5.139e-03  2.850e-03   1.803    0.0743 .
     wp            1.239e-03  8.188e-04   1.513    0.1333
685  ---
     Signif. codes:  0 '***' 0.001 '**' 0.01 '*' 0.05 '.' 0.1 ' ' 1

     Residual standard error: 0.008747 on 102 degrees of freedom
     Multiple R-squared:  0.2914, Adjusted R-squared:  0.2359
F-statistic: 5.244 on 8 and 102 DF,  p-value: 1.657e-05

     cluster4 ~ qbo + pacwarm + aao + pna + whwp + tsa + wp
     Residuals:
695       Min        1Q     Median        3Q       Max
     -0.0241264 -0.0068213  0.0008309  0.0065308  0.0214588

     Coefficients:
                 Estimate Std. Error t value Pr(>|t|)
(Intercept)  9.505e-03  2.029e-03   4.685 8.60e-06 ***
     qbo          4.643e-04  6.565e-05   7.073 1.88e-10 ***
     pacwarm     -2.963e-02  6.248e-03  -4.742 6.82e-06 ***
     aao         -4.236e-03  9.916e-04  -4.272 4.33e-05 ***
     pna         -4.256e-03  1.094e-03  -3.888 0.000179 ***
whwp         1.533e-03  5.363e-04   2.858 0.005163 **
     tsa         -6.994e-03  3.223e-03  -2.170 0.032304 *
     wp           1.517e-03  9.401e-04   1.613 0.109766
     ---
     Signif. codes:  0 '***' 0.001 '**' 0.01 '*' 0.05 '.' 0.1 ' ' 1
     Residual standard error: 0.01017 on 103 degrees of freedom
     Multiple R-squared:  0.485,  Adjusted R-squared:   0.45
     F-statistic: 13.86 on 7 and 103 DF,  p-value: 1.54e-12

     cluster5 ~ pacwarm + qbo + pna + wp + tsa + tna +
        aao + amo

     Residuals:
Min        1Q     Median        3Q       Max
     -0.068777 -0.018276   0.002122   0.019003   0.053377

     Coefficients:
                 Estimate Std. Error t value Pr(>|t|)
(Intercept)  0.0325184  0.0056697   5.736 9.99e-08 ***
     pacwarm     -0.0410161  0.0159533  -2.571  0.01158 *
     qbo          0.0009835  0.0002028   4.849 4.45e-06 ***
     pna         -0.0064672  0.0028895  -2.238  0.02739 *
     wp          -0.0069884  0.0025421  -2.749  0.00707 **
tsa         -0.0212969  0.0103254  -2.063  0.04169 *
     tna         -0.0166577  0.0083008  -2.007  0.04742 *
     aao          0.0042951  0.0028774   1.493  0.13860
     amo         -0.0282182  0.0190517  -1.481  0.14165
     ---
Signif. codes:  0 '***' 0.001 '**' 0.01 '*' 0.05 '.' 0.1 ' ' 1

     Residual standard error: 0.02682 on 102 degrees of freedom
     Multiple R-squared:  0.4204, Adjusted R-squared:  0.375
     F-statistic: 9.248 on 8 and 102 DF,  p-value: 1.658e-09
     cluster6 ~ pacwarm + tsa + qbo + ao + tna + wp +
        pdo
```



```
Residuals:
     Min       1Q    Median       3Q       Max
-0.081610 -0.015697  0.002089  0.012517  0.093024

Coefficients:
              Estimate Std. Error t value Pr(>|t|)
(Intercept)  0.0344383  0.0054041   6.373 5.31e-09 ***
pacwarm     -0.0223392  0.0174548  -1.280  0.20348
tsa         -0.0484257  0.0098959  -4.894 3.67e-06 ***
qbo          0.0008939  0.0001897   4.713 7.67e-06 ***
ao          -0.0073251  0.0024063  -3.044  0.00296 **
tna         -0.0258922  0.0084980  -3.047  0.00294 **
wp          -0.0046411  0.0024391  -1.903  0.05986 .
pdo         -0.0047650  0.0027319  -1.744  0.08411 .
---
Signif. codes:  0 '***' 0.001 '**' 0.01 '*' 0.05 '.' 0.1 ' ' 1

Residual standard error: 0.02618 on 103 degrees of freedom
Multiple R-squared:  0.4304, Adjusted R-squared:  0.3917
F-statistic: 11.12 on 7 and 103 DF,  p-value: 2.075e-10
```