# Peer review of "Signs of climate variability in double tropopause global distribution from radio occultation data"

_EGUsphere, 2024_

## Referee Comment (RC1)

**Review of 'Signs of climate variability in double tropopause global distribution from radio occultation data' by Alejandro de la Torre et al.**

There is undoubted value in the use of GNSS-RO observations to monitor and understand changes in complex tropopause characteristics including conditions of multiple tropopauses. The premise of the analysis is therefore strong. The authors are to be commended for taking this on and I would encourage them to work further on it. However, there is probably considerable work required for this to be publishable. Concerns relate to the appropriateness of the statistical approach, the lack of physical interpretation of the results in terms of fundamental processes and the overall structuring of the paper including the complete absence of a classical discussion and conclusions section.

I limit below to only major comments given the need for substantial work before this could be publishable.

**Major comments**
1. The consideration of solely double tropopauses is somewhat limiting. There are many interesting complex tropopause cases illuminated by RO and this should be at the very least acknowledged. Also, the sensitivity to the single definition of a double tropopause deployed is an obvious weakness. If you had chosen different objective criteria to define a double tropopause event how would your analysis have differed?
2. Overall paper structure is really far from the classical structure for a paper, that being introduction – methods – results – discussion-conclusion. Interleaving methods and results throughout makes for a very challenging read for a reader with new aspects of methods suddenly being dropped at random points in the text. Rewriting the paper in the more classical way would probably make for an easier read. In particular the lack of a discussion and conclusions means the 'so what' part is almost entirely missing. You need to close by placing your analysis in the broader context, highlight any caveats, and outline some potential future directions and open questions.
3. Figures in general need considerable work for clarity. In particular figure 1 is indecipherable to the reader as presented. This could instead, for example, have been presented as a stacked plot of timeseries by latitude bands N to S with the same vertical axes ranges extending vertically across a whole page enabling a reader to easily ascertain latitudinal variations. This could have avoided trying to find 18 colours which are challenging for most and indecipherable to colour-blind readers. Other figures have similar challenges but Figure 1 is by far the most challenging to comprehend as currently presented.
4. Why were the 29 indicies chosen and why do you expect these to be important in double tropopause behaviour? This married to the lack of physical interpretation is problematic. When you do compare them it currently leaves a reader with a perhaps unfortunate impression that you are proverbially throwing spaghetti at the wall in the hope that some of it sticks. I doubt this was the case but as currently written it is hard to tell on what basis you chose this set and why you think all these might, plausibly, matter. This comes to the point made in the

opening remarks that this is very statistically heavy and you really need more physical understanding in the piece as a whole.

5. In the cluster analysis work from the analysis as shown it is hard for me to really tell that there truly are six distinct clusters. In Figure 3 they just look like cuts driven by the arbitrary selection of six clusters in what is very much a continuum of behaviour with no obvious centring into distinct clusters driven by likely distinct physical behaviour. This is compounded in Figure 6 where in particular cluster 4's distribution suggests this cluster is not driven in any way by the physics with cluster placement ranging across almost all latitude bands.

6. Given significant seasonality in the latitudinal distribution of key aspects of circulation relevant to double tropopauses, the use of a seasonally varying criteria or criteria that track key features from e.g. reanalyses may have been considerably more elucidating. We know that double tropopauses are more common in key physical conditions as you have alluded to. Using a fixed lat-lon distribution when features may be repeatedly transient across such fixed grids on an annual and semi-annual basis probably explains much of the annual and higher harmonics structure in figures 4 and 5. Again, this is highlighting the need to really think about the physics here. The use of a fixed lat-lon grid vs a feature tracking approach e.g. following the sub-tropical and polar jets and the ITCZ throughout the year should be considered in revisions. A feature tracking approach which could be utilised by e.g. using ERA5 diagnostics for features of interest might give a clearer picture than your current fixed lat-lon approach.

7. The multivariate regression really needs much more physical interpretation to be of any value. At present the statistical results are presented and any physical interpretation pretty much left as an exercise for the interested reader. Statistical significance is a necessary but insufficient condition to draw robust conclusions here. It is necessary to understand physically what these results are showing us and what they mean. Why is something leading or lagging and if something is lagging does that mean that somehow double tropopauses are causing that phenomena? There is an absolute need for understanding physically what your results mean here for them to have any scientific value. I can understand how double tropopause features may lag a given phenomena, but I am unsure how to interpret a result saying they are a leading indicator. Table 1 is thus very confusing to me as a reader presently.

8. I am not really sure how I should interpret figure 9 as presented. In particular in clusters 1 and 3 the test RMSE is consistently lower than the training RMSE which makes no logical sense. This may highlight that the cluster definition is not appropriate (see earlier point) and that the behaviour within clusters is non-stationary in interesting ways as a result.

9. Table 2 again you are making the reader do the lifting of the physical understanding as to why these particular modes might matter to these particular clusters. Taken together with Table 1 I have a real challenge thinking how to interpret your results here. You need to help a reader understand how to interpret these combined results.

10. I am unclear why so much of what would nominally be considered key results is left in the supplement and not discussed at all in the main text. I may have

missed it but I failed to note a reference to it and certainly a substantive analysis and discussion of these results.

11. I am always loathed as a native English speaker to make this point as I could never even attempt to write a paper in any language other than English let alone to such as a standard, but the paper overall is a heavy read and either getting a native English proof reader or engaging a native English speaking co-author to help in the rewrite and restructure would be helpful.

---

## Referee Comment (RC2)

**Review of "Signs of climate variability in double tropopause global distribution from radio occultation data" by Alejandro de la Torre et al.**

The authors present a complex statistical methodology for the analysis of double tropopause (DT) variability. The DT topic is not extensively researched, and the presented methodology has potential to provide valuable results, so the general idea of the study is a welcome one. However, especially in section 3, my impression is that not enough testing and optimization of the method has been performed, which limits the result's performance and interpretability in later sections.

The statistical side of the work is presented in a very detailed way. However the discussion of results regarding tropopause dynamics and previous publications is nearly absent from the manuscript. Also I have concerns about the setup of the clustering analysis which serves as the base for the multivariate linear regression later. All together, I feel the present manuscript is still a long way from being fit for publication.

Although my recommendation is 'reject' for the manuscript in its current form, as I state above the method has good potential and I would encourage resubmission once the extensive list of issues is rigorously addressed.

\#
\# **Major comments**
\#

**M1.1**: There's a lack of relation and discussion of processes responsible for DT, or DT types, with the clusters found in your analysis.

**M1.2**: The clustering analysis is done with very basic parameters, I am not sure they are the best choice.
→ Using standard deviation of NDT' blends all modes and timescales of variability together, this compounds with my comment M1.1 and makes interpretation of the results very, very difficult, if one wants to go further than just showing the statistics.
→ See my "**general comment on section 3**" for suggestions on this issue.

**M2:** Sometimes I noticed the authors do not justify some parameter choices in their analysis, simply stating that the result is 'reasonable', which is not sufficient in my view.
→ In other places, the authors present some results in an exaggerately positive way, while to me they are unconvincing or even cause for concern in one instance.
→ I marked the most glaring examples of this with "**(M2)**" in the individual comments below

**M3:** No interpretation/discussion is provided, whatsoever, for results in section 3 in terms of STE, atmospheric dynamics or previous DT literature. For sections 4-6, this is limited to a couple of paragraphs at most.
→ On the other hand, explanations on some statistical methods are overly long and include sometimes unnecessary definitions of standard statistical measures.

Please find individual comments and suggestions for each section below:

#
**Abstract**
#

2nd half of the abstract has too much technical jargon on cluster analysis and linear fitting.

No main results or conclusions are highlighted. E.g.:
*"the most relevant climatic indices for the distribution of NDT' are identified."* → Which ones?!

Whereas the authors state that the main focus of the manuscript is the methodological approach, main results and potential applicability should be highlighted from the beginning.

#
**Introduction**
#

l. 57-58: *"and detected in cloud-top inversion layers (Biondi et al., 2012)"* → shouldn't this be considered as an artifact of the lapse-rate definition of multiple tropopauses?

l. 71-74: feels very vague, and big data are not used anywhere in your manuscript.

#
**Sect. 2**
#

l. 106: **Foelsche et al. (2011)** missing from reference list, same with **Angerer et al. (2017).** Please check throughout manuscript for missing items in reference list.

l. 109: why do you start at 2006 and not 2001? Should state the step-change in RO data amount from the start of COSMIC

l. 110: DT percentage or frequency has been shown in previous studies on multiple tropopauses, see e.g. references from section 4.1 in **Wilhelmsen et al. (2020).** Please cite them along, the most relevant ones.
l. 110: also, Wilhelmsen didn't invent the lapse-rate tropopause definition, please cite the WMO definition that the algorithm uses.

l. 110: perhaps this is described in the Wilhelmsen or Angerer studies, but what happens with the RO profiles where the tropopause cannot be found? I know from experience there's always a small percentage of those, please give some number on the discarded profiles.

l. 115: Wilhelmsen used 5x5 degrees, so this difference should be pointed out since you don't follow their horizontal resolution.

→ Please don't use the label N2/N1 in figure labels. "DT frequency" or $DT_{freq}$ is much more reader-friendly and intuitive than N2/N1.
I even kindly suggest to use $DT_{freq}$ as a substitute for NDT throughout the manuscript.

l. 121: *"complex pattern with a prevailing temporal variability that depends essentially on the latitude"* → can't really say anything from Fig. 1 in the current form, and of course, there are different variability modes at different latitude bands.

Figure 1: not publication-worthy
→ Please substitute the y-axis for Latitude, and plot DT frequency as color shading, this is the proper way of visualizing the same information.
→ Also format month number into "YYYY", perhaps label every two years.

#
**Sect. 3**
#

l. 139-140: *"The mean values of the NDT time series and the standard deviations of the NDT' time series are then used for the clustering."*
→ How do the authors justify these settings, why are these the optimal variables for clustering?
→ Also, the authors should describe a bit what these variables represent e.g. in terms of STE, otherwise for many readers this may seem as just a statistical exercise with the most basic parameters of the NDT distribution.

l. 165: about the cutoff distance of 0.07, perhaps some sensitivity experiments with +- 5% or 10% of that value would be reassuring, if shown in a supplement.
Also, please state in this paragraph, what is the usual range of cutoff distances in CA in general.
**(M2)**

Fig. 3 caption: Does each cluster have the same color in Figs. 2 and 3? Please specify this.

State somewhere in the text at the beginning of section 3 that the 'arbitrary' color scheme from Fig. 2 will be the same in all plots.

l. 175: *"a reasonable separation of objects is achieved"*.
→ This judgment is based on what property of the plot?
→ Also I disagree with the statement, Fig. 3 rather looks like a continuous scatterplot.
**(M2)**

Fig. 3: → Please show the corresponding probability density estimates of this scatterplot, maybe there are relative density maxima corresponding to some clusters. But from the current plot one can't say.

Figs. 2, 4, 5, and 6: I strongly suggest to number the clusters according to the distributions in Fig. 3, from left to right. In Fig. 2, please add the corresponding cluster numbers which are missing.

Fig. 6 and corresponding text:
→ Discussion missing, please at least discuss the different clusters in relation to different high DT frequency regions from e.g. **Wilhelmsen et al. (2020).**

→ In my opinion, describing the clusters as symmetrically distributed relative to the equator has little meaning: they are related to the subtropical jets on both hemispheres, so sure this will look somewhat symmetric, but it's the relation to the jets that has interpretative value.
→ Fig. 6 looks quite coarse, since it works with monthly timeseries, a refinement to 5x5 degree or better is possible from RO coverage, and would be very welcome.

l. 204-205: *"The interconnected nature of each sub-region is highlighted in the polar, sub-polar and equatorial regions by clusters 1 and 4."*
**(M2)**
→ I'm sorry to put it this way, but to me this sentence is quite euphemistic. What I see is the method's weakness: equatorial and polar DT's have very little in common, yet the clustering method is mixing them up.
→ Clusters 1 and 4 are next to each other in Fig. 3, and their spatial distribution in Fig. 6 has no distinct structure, it makes me doubt about the method's ability to separate some DT features – with the used settings in this manuscript. See below for further suggestions on the method.

**General comment on section 3:**
→ I am not sure the parameters used for the clustering analysis are the most meaningful. With the method as is, all DT types as well as all modes of variability are blended together and really difficult to separate. I have a couple of suggestions that should help with the clustering and interpretation of results:
        - Separating NDT' by time-scales, e.g. into subseasonal, interannual, QBO-specific, ENSO-specific… would make any cluster regions found easier to interpret, and relatable to previous works. For example, a good test of the clustering method would be to compare its output regions to the El Nino – La Nina differences shown in **Wilhelmsen et al. (2020)**, their Fig. 3c.
        - Combine with different parameters for clustering: e.g. DT depth (difference in height between the two tropopauses) or its standard deviation could be tested instead of std(NDT') to see whether the clustering performance improves. DT depth properties should distinguish equatorial and mid-latitude / polar DT much better than DT frequency alone.

→ I think some sensitivity tests with other parameters and/or case-studies (e.g. ENSO-specific timescale) are necessary in order to reassure the audience that this method can give some useful and meaningful output, that can be related and build upon previous DT studies. Especially the current Figure 6, in my opinion is rather far from being convincing on the method's potential, and it's done with a single setting for clustering on, again, basic distribution parameters of frequency.
→ Once clustering gives something more robust, one may expect a lot more juice coming from the multivariate regression analyses.

#
**Sect. 4+5**
#

I feel these sections could be summarized into a couple of methods subsections. They are very heavy in the present manuscript, their extended version with all the minutiae can be moved to an appendix or supplement.

Same with the first half of section 6 actually.

l. 320-345: I don't think it's necessary to explain what is the $R^2$, adjusted $R^2$ and F-statistic, these are standard things… please explain the most relevant details for model evaluation in a very summarized way.

#
**Sect. 6**
#

l. 379-385: I was thinking exactly this while reading everything after section 3: the clustering analysis feels to me like the bottleneck for the rest of the methods/results – it is imperative that the results in section 3 are robust and thoroughly tested, then sections 4-6 will benefit greatly and interpretation beyond just the statistics will be more straightforward.

l. 387-403: this is the only paragraph where the results from the multivariate linear regression are discussed, this part should be markedly extended.
→ From these results, please highlight what is added upon the referenced work.
→ From the results of this section, can you say something about which climate indices affect STE the most? Can your methodology provide a way forward to answer such question?

l. 430-435: I don't think it's necessary to explain what a q-q plot is…

Figs. 10-11:
→ Both appear a bit pixelated, please increase quality, and reduce the horizontal separation between the red boxes. Also, titles are repeated in each sub-panel, you can save a lot of space using one title above all panels.

l. 439-445: this can be considered an outlook, please create a separate section 'Summary & Outlook' or similar that summarizes main results and discusses possible applications and adaptability of your method.

**Appendix C**: is it really necessary that these very technical infos stay within the main manuscript? I'd suggest to move it to a separate supplement document.

**Data availability statement:** state here (or reference in methods section) what software is used for clustering analysis and regression model.

---

## Author Comment (AC1)

**Review of 'Signs of climate variability in double tropopause global distribution from radio occultation data' by Alejandro de la Torre et al.**

There is undoubted value in the use of GNSS-RO observations to monitor and understand changes in complex tropopause characteristics including conditions of multiple tropopauses. The premise of the analysis is therefore strong. The authors are to be commended for taking this on and I would encourage them to work further on it.

However, there is probably considerable work required for this to be publishable. Concerns relate to the appropriateness of the statistical approach, the lack of physical interpretation of the results in terms of fundamental processes and the overall structuring of the paper including the complete absence of a classical discussion and conclusions section.

I limit below to only major comments given the need for substantial work before this could be publishable.

**Major comments**

1. The consideration of solely double tropopauses is somewhat limiting. There are many interesting complex tropopause cases illuminated by RO and this should be at the very least acknowledged. Also, the sensitivity to the single definition of a double tropopause deployed is an obvious weakness. If you had chosen different objective criteria to define a double tropopause event how would your analysis have differed?

*Thank you for this comment. This study focuses clearly on the relation (and correlation) of double tropopause occurrence and climate indices. Of course, over the last two decades there were several complex tropopause studies based on RO data. Because of the properties of the RO technique tropopauses and double tropopauses can be detected precisely on a global scale. Several of these previous studies describe climatologies and even trends in tropopause parameters. Some of these studies are already listed in the references. In the revised version, we will include a broader spectrum of this previous publications related to tropopause and double tropopause investigations using RO or other datasets.*

*In this study we use the WMO definition of the lapse-rate tropopause (Wilhelmsen et al 2020). This definition includes also the conditions to detect double tropopauses. Of course, the WMO definition from 1957 was developed based on datasets with a coarse vertical resolution.*

*Due to the availability of high vertical resolution datasets (radiosondes and, e.g., RO data) some modifications (in comparison to the pure WMO definition) on the tropopause detection retrievals have been performed, e.g., Schmidt et al. (2005) and Birner (2006) (see below).*

*But, if you compare (double) tropopause climatologies from different authors (that usually avoid giving precise information on the tropopause detection algorithms) the climatologies are very similar, i.e. the results of our double tropopause climatology are robust.*

*In summary, we would argue that even if there are small differences in the tropopause algorithms the general picture of the tropopause climatologies is the same. From that we further conclude that our results based on our analysis would have no basic differences if we had chosen a (small) different criterion to define the double tropopause.*

*Moreover, in the revised version we include the following additional DT studies and the corresponding*

*main focus in each of them, in addition to the already referenced:*

*Randel, W. J., D. J. Seidel, and L. L. Pan (2007), Observational characteristics of double tropopauses, J. Geophys. Res., 112, D07309, doi:10.1029/2006JD007904.*

*Temperature profiles in the extratropics often exhibit multiple tropopauses (as defined using the lapse rate definition). In this work the authors studied the observational characteristics of DT based on radiosondes, ERA40 reanalysis, and GPS radio occultation temperature profiles.*

*Schmidt, T., J.-P. Cammas, H. G. J. Smit, S. Heise, J. Wickert, and A. Haser (2010), Observational characteristics of the tropopause inversion layer derived from CHAMP/GRACE radio occultations and MOZAIC aircraft data, J. Geophys. Res., 115, D24304, doi:10.1029/2010JD014284.*

*The characteristics of the Northern Hemisphere (NH) midlatitude (40°N–60°N) tropopause inversion layer (TIL) based on two data sets. First, temperature measurements from GPS radio occultation data (CHAMP and GRACE) for the time interval 2001–2009 are used to exhibit seasonal properties of the TIL. Secondly, high-resolution temperature and trace gas profile measurements on board commercial aircrafts (Measurement of Ozone and Water Vapor by Airbus In-Service Aircraft (MOZAIC) program) from 2001–2008 for the NH midlatitude (40°N–60°N) region are used to characterize the TIL as a mixing layer around the tropopause.*

*Castanheira et al. (2012), Relationships between Brewer-Dobson circulation, double tropopauses, ozone and stratospheric water vapour. Atmospheric Chemistry and Physics.10.5194/acp-12-10195-2012.2012.*

*Statistical relationships between the variability of the area covered by DT events, the strength of the tropical upwelling, the total column ozone and of the lower stratospheric water vapour are analyzed. The analysis is based on both reanalysed data (ERA-Interim) and HIRDLS satellite data.*

*Liu, C., & Barnes, E. A. (2018), Synoptic formation of double tropopauses. Journal of Geophysical Research: Atmospheres, 123, 693–707. https://doi.org/10.1002/2017JD027941*

*As DT are ubiquitous in the midlatitude winter hemisphere and represent the vertical stacking of two stable tropopause layers separated by a less stable layer, by analyzing COSMIC GPS data, reanalysis, and eddy life cycle simulations, the authors demonstrate that they often occur during Rossby wave breaking and act to increase the stratosphere-to-troposphere exchange of mass. Moreover, the adiabatic formation of double tropopauses and two possible mechanisms by which they can occur were proposed.*

*Shao, J., Zhang, J., Tian, Y., Wang, W., Huang, K., & Zhang, S. (2023), Tropospheric gravity waves increase the likelihood of double tropopauses. Geophysical Research Letters, 50, e2023GL105724. https://doi.org/10.1029/2023GL105724.*

*As the tropopause region is crucial for the stratosphere-troposphere exchange (STE) and acts as an indicator of climate change, DT events act to increase the STE process but their driving mechanisms remain an open question. The present assessment offers for the first time the linkage between tropospheric gravity waves and DT events by exploring a global data set of multi-year radiosonde measurements.*

*Schmidt et al. (2005), GPS radio occultation with CHAMP and SAC-C: global monitoring of thermal tropopause parameters. Atmospheric Chemistry and Physics.10.5194/acp-5-1473-2005.*

*Birner, T. (2006), Fine-scale structure of the extratropical tropopause region, J. Geophys. Res., 111, D04104, doi:10.1029/2005JD006301.*

*We have included an additional discussion regarding tropopause dynamics in sections 1 and 2.1 and in 2.2.1 the proposed cluster analysis is discussed in more detail.*

2. Overall paper structure is really far from the classical structure for a paper, that being introduction – methods – results – discussion-conclusion. Interleaving methods and results throughout makes for a very challenging read for a reader with new aspects of methods suddenly being dropped at random points in the text. Rewriting the paper in the more classical way would probably make for an easier read. In particular the lack of a discussion and conclusions means the 'so what' part is almost entirely missing. You need to close by placing your analysis in the broader context, highlight any caveats, and outline some potential future directions and open questions.

*In the revised version, which was re-written following a completely different structure (Abstract, Introduction, Data and Methodology, Results, Discussion and Conclusions), starting from a DT database obtained from RO observations, we propose to explore a possible relationship between the spatio-temporal distribution of DTs and a set of monthly climate indices, with a primary focus on the methodological approach. With the main purpose to illustrate this idea, we first apply a cluster analysis to geographically associate DT occurrences. Secondly, we construct a multivariate linear regression using a progression of different models, considering train and test populations, to identify climate indices relevant for DT occurrence. Then, these preliminary results should be considered as the beginning of a more in-depth analysis, currently in progress, in which the robustness of the results is still pending to be found and established.*

3. Figures in general need considerable work for clarity. In particular figure 1 is indecipherable to the reader as presented. This could instead, for example, have been presented as a stacked plot of timeseries by latitude bands N to S with the same vertical axes ranges extending vertically across a whole page enabling a reader to easily ascertain latitudinal variations. This could have avoided trying to find 18 colours which are challenging for most and indecipherable to colour- blind readers. Other figures have similar challenges but Figure 1 is by far the most challenging to comprehend as currently presented.

*We agree with this comment. In the revised version, Figure 1 was eliminated, as it is not essential to illustrate our results and we tried to improve the resolution of some of the remaining figures.*

4. Why were the 29 indicies chosen and why do you expect these to be important in double tropopause behaviour? This married to the lack of physical interpretation is problematic. When you do compare them it currently leaves a reader with a perhaps unfortunate impression that you are proverbially throwing spaghetti at the wall in the hope that some of it sticks. I doubt this was the case but as currently written it is hard to tell on what basis you chose this set and why you think all these might, plausibly, matter. This comes to the point made in the

opening remarks that this is very statistically heavy and you really need more physical understanding in the piece as a whole.

*Climatic indices play a crucial role in understanding the general circulation of the atmosphere by providing valuable insights into climate patterns and variability, climate change and in the link between the ocean and the atmosphere. Moreover, for improving our understanding of the interconnected nature of Earth's climate system. Overall, climatic indices are essential tools for meteorologists, climatologists, and policymakers in understanding and responding to*

*atmospheric dynamics. Besides, double tropopauses are produced due to specific atmospheric conditions that lead to the formation of distinct tropopause layers as a consequence of different dynamic or thermodynamic situations, i.e., stratospheric temperature inversions, vertical shear and stability, convective activity and jet streams. These comments were included in section 2.1.*

5. In the cluster analysis work from the analysis as shown it is hard for me to really tell that there truly are six distinct clusters. In Figure 3 they just look like cuts driven by the arbitrary selection of six clusters in what is very much a continuum of behaviour with no obvious centering into distinct clusters driven by likely distinct physical behaviour. This is compounded in Figure 6 where in particular cluster 4's distribution suggests this cluster is not driven in any way by the physics with cluster placement ranging across almost all latitude bands.

*The non-hierarchical K-means cluster analysis only follows the classification indicated above by the hierarchical method into 6 groups. Additional parameters of higher order than the mean NDT and the standard deviation of NDT' could also have been included. This is one of the main assumptions of our analysis, resulting in a well-defined object separation in Figure 2 of the new version. We do not expect that final physical conclusions can be obtained before the robustness of the results (time series classification and model relating NTD' to climate indices) is guaranteed.*

6. Given significant seasonality in the latitudinal distribution of key aspects of circulation relevant to double tropopauses, the use of a seasonally varying criteria or criteria that track key features from e.g. reanalyses may have been considerably more elucidating. We know that double tropopauses are more common in key physical conditions as you have alluded to. Using a fixed lat-lon distribution when features may be repeatedly transient across such fixed grids on an annual and semi-annual basis probably explains much of the annual and higher harmonics structure in figures 4 and 5. Again, this is highlighting the need to really think about the physics here. The use of a fixed lat-lon grid vs a feature tracking approach e.g. following the sub-tropical and polar jets and the ITCZ throughout the year should be considered in revisions. A feature tracking approach which could be utilized by e.g. using ERA5 diagnostics for features of interest might give a clearer picture than your current fixed lat-lon approach.

*We are aware that climate studies often use latitude and longitude grids to represent global data, but alternative methods have emerged, especially with machine learning. Several approaches have been developed to model and analyze global climate without relying on traditional grid clustering. Some of these methods focus on pattern recognition, dimensionality reduction, and leveraging irregular data inputs: graph neural networks, spectral methods and harmonic analysis, gaussian process models, unsupervised learning with autoencoders or variational autoencoders and self-organizing maps, temporal and spatial attention mechanisms in transformers and principal component analysis. (We postpone a clustering of global climate data based on large-scale patterns rather than latitude and longitude grids for a next contribution).*

*On the other hand, to describe the behavior of a global variable in terms of climate indices states, rather than grouping regions, we can use a few advanced methods to analyze how global variables like DTfrequency, temperature, precipitation, or wind speed are influenced by climate indices such as the ENSO, NAO, or PDO: Multivariate regression models with climate indices, state-space models, dynamic mode decomposition with climate indices, canonical correlation analysis and machine learning methods like random forests or neural networks (we begun here with the first of these methods). These comments are included in section 4.*

7. The multivariate regression really needs much more physical interpretation to be of any value. At present the statistical results are presented and any physical interpretation pretty much left as an exercise for the interested reader. Statistical significance is a necessary but insufficient condition to draw robust conclusions here. It is necessary to understand physically what these results are showing us and what they mean. Why is something leading or lagging and if something is lagging does that mean that somehow double tropopauses are causing that phenomena? There is an absolute need for understanding physically what your results mean here for them to have any scientific value. I can understand how double tropopause features may lag a given phenomena, but I am unsure how to interpret a result saying they are a leading indicator. Table 1 is thus very confusing to me as a reader presently.

*As mentioned above, the results above presented must be strictly considered as a first step of a deep analysis to reveal the model that minimizes RMSE in test and training data. This presentation should be considered as the beginning of a more in-depth analysis, currently in progress, in which the robustness of the results can be verified. Prior to the development of any model, as NDT´ and the features may present the best cross-correlation for time lags k different from zero, it is worth considering k values within an interval around k = 0. A resulting k ≠ 0 value may indicate the ability of NDT´ to anticipate a given feature, or vice versa. Moreover, a significant maximum CC may indicate the possible relative relevance of the respective feature in relation to the others (Section 2.2.2 and 4).*

8. I am not really sure how I should interpret figure 9 as presented. In particular in clusters 1 and 3 the test RMSE is consistently lower than the training RMSE which makes no logical sense. This may highlight that the cluster definition is not appropriate (see earlier point) and that the behaviour within clusters is non- stationary in interesting ways as a result.

*RMSE is shown for the training and test samples. This is an indication of the degree of possible model overfitting ($RMSE_{test} >> RMSE_{train}$). A possible indication of overfitting appears in clusters 2, 4 and 5. In clusters 1, 3 and 6, $RMSE_{test} << RMSE_{train}$ suggests that the model performs better in the test population, as desired. We recall that if the test data are similar to the training data but without as much noise, the model may perform better in the test population, also resulting in a lower RMSE. Moreover, if the training data contain many outliers that negatively affect the model, and those outliers are not present in the test data, the RMSE also may be higher in the training data than in the test data (section 3.2.2).*

9. Table 2 again you are making the reader do the lifting of the physical understanding as to why these particular modes might matter to these particular clusters. Taken together with Table 1 I have a real challenge thinking how to interpret your results here. You need to help a reader understand how to interpret these combined results.

*Table 2 lists the features selected by the AIC forward step-wise method, it is to say, which features are significant or relevant according to the best multivariate regression "with ALL" model found. $R^2$, adjusted $R^2$ and F-statistic values in each cluster are included too. Previously to the model, in Table 1 we indicate the lag corresponding to the best CC found between each of the features and NDT', with k ranging from -5 to + 5 months. The relative enhanced significance of each feature and the possibility that each feature anticipates or delays NDT´ by $k_0$ months is highlighted.*

10. I am unclear why so much of what would nominally be considered key results is left in the supplement and not discussed at all in the main text. I may have missed it but I failed to note a reference to it and certainly a substantive analysis and discussion of these results.

    *Sections 3 and 4 include further explanations of the material contained in appendices C and D. In this last, for additional information the coefficients and the features corresponding to the "with ALL" model, in clusters 1 to 6, are included. We believe that the definition of the meaning of each parameter is not necessary.*

11. I am always loathed as a native English speaker to make this point as I could never even attempt to write a paper in any language other than English let alone to such as a standard, but the paper overall is a heavy read and either getting a native English proof reader or engaging a native English speaking co-author to help in the rewrite and restructure would be helpful.

    *In the new version we have enlisted the help of an English proofreader.*

---

## Author Comment (AC2)

**Review of "Signs of climate variability in double tropopause global distributionfrom radio occultation data" by Alejandro de la Torre et al.**

The authors present a complex statistical methodology for the analysis of double tropopause (DT) variability. The DT topic is not extensively researched, and the presented methodology has potentialto provide valuable results, so the general idea of the study is a welcome one. However, especially in section 3, my impression is that not enough testing and optimization of the method has been performed, which limits the result's performance and interpretability in later sections.

*Thank you for your comments. We agree with this comment. In the revised version, which was re-written following a completely different structure (Abstract, Introduction, Data and Methodology, Results, Discussion and Conclusions), starting from a DT database obtained from RO observations, we propose to explore a possible relationship between the spatio-temporal distribution of DTs and a set of monthly climate indices, with a primary focus on the methodological approach. With the main purpose to illustrate this idea, we first apply a cluster analysis to geographically associate DT occurrences. Secondly, we construct a multivariate linear regression using a progression of different models, considering train and test populations, to identify climate indices relevant for DT occurrence. Then, these preliminary results should be considered as the beginning of a more in-depth analysis, currently in progress, in which the robustness of the results is still pending to be found and established.*

The statistical side of the work is presented in a very detailed way. However the discussion of results regarding tropopause dynamics and previous publications is nearly absent from the manuscript. Also I have concerns about the setup of the clustering analysis which serves as the base for the multivariate linear regression later. All together, I feel the present manuscript is still a long way from being fit for publication.

> *We have included an additional discussion regarding tropopause dynamics in sections 1 and 2.1 and in 2.2.1 the proposed cluster analysis is discussed in more detail.*

> Although my recommendation is 'reject' for the manuscript in its current form, as I state above themethod has good potential and I would encourage resubmission once the extensive list of issues isrigorously addressed.

**# Major comments**

> **M1.1**: There's a lack of relation and discussion of processes responsible for DT, or DT types, with the clusters found in your analysis.

*This study focuses on the relation (and correlation) of double tropopause occurrence and climate indices. Of course, over the last two decades there were several complex tropopause studies based on RO data. Because of the properties of the RO technique tropopauses and double tropopauses can be detected precisely on a global scale. Several of these previous studies describe climatologies and even trends in tropopause parameters. Some of these studies are already listed in the references. In the revised version, we will include a broader spectrum of this previous publications related to tropopause and double tropopause investigations using RO or other datasets.*

*In this study we use the WMO definition of the lapse-rate tropopause (Wilhelmsen et al 2020). This definition includes also the conditions to detect double tropopauses. Of course, the WMO definition from 1957 was developed based on datasets with a coarse vertical resolution.*

*Due to the availability of high vertical resolution datasets (radiosondes and, e.g., RO data) some modifications (in comparison to the pure WMO definition) on the tropopause detection retrievals have been performed, e.g., Schmidt et al. (2005) and Birner (2006) (see below).*

*But, if you compare (double) tropopause climatologies from different authors (that usually avoid giving precise information on the tropopause detection algorithms) the climatologies are very similar, i.e. the results of our double tropopause climatology are robust.*

*In summary, we would argue that even if there are small differences in the tropopause algorithms the general picture of the tropopause climatologies is the same. From that we further conclude that our results based on our analysis would have no basic differences if we had chosen a (small) different criterion to define the double tropopause.*

*Moreover, in the revised version we include the following additional DT studies and the corresponding main focus in each of them, in addition to the already referenced:*

*Randel, W. J., D. J. Seidel, and L. L. Pan (2007), Observational characteristics of double tropopauses, J. Geophys. Res., 112, D07309, doi:10.1029/2006JD007904.*

*Temperature profiles in the extratropics often exhibit multiple tropopauses (as defined using the lapse rate definition). In this work the authors studied the observational characteristics of DT based on radiosondes, ERA40 reanalysis, and GPS radio occultation temperature profiles.*

*Schmidt, T., J.-P. Cammas, H. G. J. Smit, S. Heise, J. Wickert, and A. Haser (2010), Observational characteristics of the tropopause inversion layer derived from CHAMP/GRACE radio occultations and MOZAIC aircraft data, J. Geophys. Res., 115, D24304, doi:10.1029/2010JD014284.*

*The characteristics of the Northern Hemisphere (NH) midlatitude (40°N–60°N) tropopause inversion layer (TIL) based on two data sets. First, temperature measurements from GPS radio occultation data (CHAMP and GRACE) for the time interval 2001–2009 are used to exhibit seasonal properties of the TIL. Secondly, high-resolution temperature and trace gas profile measurements on board commercial aircrafts (Measurement of Ozone and Water Vapor by Airbus In-Service Aircraft (MOZAIC) program) from 2001–2008 for the NH midlatitude (40°N–60°N) region are used to characterize the TIL as a mixing layer around the tropopause.*

*Castanheira et al. (2012), Relationships between Brewer-Dobson circulation, double tropopauses, ozone and stratospheric water vapour. Atmospheric Chemistry and Physics.10.5194/acp-12-10195-2012.2012.*

*Statistical relationships between the variability of the area covered by DT events, the strength of the tropical upwelling, the total column ozone and of the lower stratospheric water vapour are analyzed. The analysis is based on both reanalysed data (ERA-Interim) and HIRDLS satellite data.*

*Liu, C., & Barnes, E. A. (2018), Synoptic formation of double tropopauses. Journal of Geophysical Research: Atmospheres, 123, 693–707.* [https://doi.org/10.1002/2017JD027941](https://doi.org/10.1002/2017JD027941)

*As DT are ubiquitous in the midlatitude winter hemisphere and represent the vertical stacking of two stable tropopause layers separated by a less stable layer, by analyzing COSMIC GPS data, reanalysis, and eddy life cycle simulations, the authors demonstrate that they often occur during Rossby wave breaking and act to increase the stratosphere-to-troposphere exchange of mass. Moreover, the adiabatic formation of double tropopauses and two possible mechanisms by which they can occur were proposed.*

*Shao, J., Zhang, J., Tian, Y., Wang, W., Huang, K., & Zhang, S. (2023), Tropospheric gravity waves increase the likelihood of double tropopauses. Geophysical Research Letters, 50, e2023GL105724.* [https://doi.org/10.1029/2023GL105724](https://doi.org/10.1029/2023GL105724)

*As the tropopause region is crucial for the stratosphere-troposphere exchange (STE) and acts as an indicator of climate change, DT events act to increase the STE process but their driving mechanisms remain an open question. The present assessment offers for the first time the linkage between tropospheric gravity waves and DT events by exploring a global data set of multi-year radiosonde measurements.*

*Schmidt et al. (2005), GPS radio occultation with CHAMP and SAC-C: global monitoring of thermal tropopause parameters. Atmospheric Chemistry and Physics.10.5194/acp-5-1473-2005.*

*Birner, T. (2006), Fine-scale structure of the extratropical tropopause region, J. Geophys. Res., 111, D04104, doi:10.1029/2005JD006301.*

**M1.2**: The clustering analysis is done with very basic parameters, I am not sure they are the best choice.

Using standard deviation of NDT' blends all modes and timescales of variability together, thiscompounds

with my comment M1.1 and makes interpretation of the results very, very difficult, ifone wants to go further than just showing the statistics.

See my "**general comment on section 3**" for suggestions on this issue.

*The choice of location and spread in terms of the mean and standard deviation constitutes a consistent start in a cluster analysis. But of course, we agree that a deeper analysis, which is not considered here, could add higher order moments to explain the symmetry of NDT'.*

**M2:** Sometimes I noticed the authors do not justify some parameter choices in their analysis, simply stating that the result is 'reasonable', which is not sufficient in my view.

In other places, the authors present some results in an exaggerately positive way, while to me they are unconvincing or even cause for concern in one instance.

I marked the most glaring examples of this with "**(M2)**" in the individual comments bel*ow*

*We aimed to replace the word "reasonable" by consistent arguments in the new version. Throughout the new text, we have attempted to balance our preliminary conclusions and assumptions.*

**M3:** No interpretation/discussion is provided, whatsoever, for results in section 3 in terms of STE, atmospheric dynamics or previous DT literature. For sections 4-6, this is limited to a couple of paragraphs at most.

On the other hand, explanations on some statistical methods are overly long and includesometimes unnecessary definitions of standard statistical measures.

*The revised version, with a different structure from the first version, includes several explanatory paragraphs added in the last two sections. We hope that the reading and comprehension of the text, together with a revision of the language, will now be more agile and clearer.*

Please find individual comments and suggestions for each section below

**Abstract**

2nd half of the abstract has too much technical jargon on cluster analysis and linear fitting.No main results or conclusions are highlighted. E.g.:

*"the most relevant climatic indices for the distribution of NDT' are identified."* → Which ones?!

Whereas the authors state that the main focus of the manuscript is the methodological approach, main results and potential applicability should be highlighted from the beginning.

*The abstract has been rewritten in the new version. Main results are now more detailed.*

**Introduction**

l. 57-58: *"and detected in cloud-top inversion layers (Biondi et al., 2012)"* → shouldn't this be considered as an artifact of the lapse-rate definition of multiple tropopauses?

*This is a possibility, please see above our discussion regarding definition of DT.*

l. 71-74: feels very vague, and big data are not used anywhere in your manuscript.

*We agree, both sentences that mention big data analysis were eliminated.*

**Sect. 2**

l. 106: **Foelsche et al. (2011)** missing from reference list, same with **Angerer et al. (2017).** Please check throughout manuscript for missing items in reference list.

*Citations included.*

l. 109: why do you start at 2006 and not 2001? Should state the step-change in RO data amountfrom the start of COSMIC

*At the beginning of the COSMIC mission, all the satellites were grouped together to ensure a synchronized start for their measurements and to simplify the deployment process (around May 2006). The final arrangement of the COSMIC satellites allowed them to cover the entire globe. Between 2001 and 2006, the density of available RO is considerably lower.*

l. 110: DT percentage or frequency has been shown in previous studies on multiple tropopauses, see e.g. references from section 4.1 in **Wilhelmsen et al. (2020).** Please cite them along, the mostrelevant ones.

l. 110: also, Wilhelmsen didn't invent the lapse-rate tropopause definition, please cite the WMOdefinition that the algorithm uses.

*In the revised version we included additional references and a discussion about the lapse-rate tropopause definition is included too (Section 2.1).*

l. 110: perhaps this is described in the Wilhelmsen or Angerer studies, but what happens with the RO profiles where the tropopause cannot be found? I know from experience there's always a small percentage of those, please give some number on the discarded profiles.

*We are not able to indicate the number or pencentage of them, but we agree that it is relatively considerably low.*

l. 115: Wilhelmsen used 5x5 degrees, so this difference should be pointed out since you don't follow their horizontal resolution.

We mention this reference in Section 2.1 and we explain that we prioritize the availability of a sufficient number of events in each cell

Please don't use the label N2/N1 in figure labels. "DT frequency" or DT**freq** is much more reader-friendly and intuitive than N2/N1.

I even kindly suggest to use DT**freq** as a substitute for NDT throughout the manuscript.

*Both suggestions were included in the revised manuscript*

l. 121: *"complex pattern with a prevailing temporal variability that depends essentially on the latitude"* →can't really say anything from Fig. 1 in the current form, and of course, there are different variability modes at different latitude bands.

Figure 1: not publication-worthy

Please substitute the y-axis for Latitude, and plot DT frequency as color shading, this is the proper way of visualizing the same information.

lso format month number into "YYYY", perhaps label every two years.

*According to comments made by both Reviewers, the "old" Figure 1 was eliminated in the revised version of the manuscript.*

**# Sect. 3#**

l. 139-140: *"The mean values of the NDT time series and the standard deviations of the NDT' time series are then used for the clustering."*

How do the authors justify these settings, why are these the optimal variables for clustering?

Also, the authors should describe a bit what these variables represent e.g. in terms of STE, otherwise for many readers this may seem as just a statistical exercise with the most basic parameters of the NDT distribution.

*The clustering is defined from two measures of location and spread of the time series: The mean values of the $DT_{freq}$ time series and the standard deviations of the $DT'_{freq}$ Additional moments of higher order could have been included too, but we feel that for a preliminary luster analysis it is advisable to consider the first two measures. A deeper analysis, which is not considered here, should progressively include higher order moments to check the symmetry of the $DT_{freq}$ distribution. The choice of mean $DT'_{freq}$ values instead of the mean $DT_{freq}$ values would not provide any information.*

l. 165: about the cutoff distance of 0.07, perhaps some sensitivity experiments with +- 5% or 10% of that value would be reassuring, if shown in a supplement

*We modified this paragraph: "In Fig. 1, within the interval [.065, .075] the resulting number of clusters is 6, so we chose a mean cutoff point equal to .07 in hierarchical clustering. This value indicates a commitment number of 6 clusters, retaining a significant number of individual cells in each cluster. We then proceed with the cluster analysis according to this classification in 6 groups."*

Also, please state in this paragraph, what is the usual range of cutoff distances in CA in general.

*If we properly understood this comment, there is not a defined range of cutoff distances in CA*

**(M2)**

Fig. 3 caption: Does each cluster have the same color in Figs. 2 and 3? Please specify this.

*No, the colors in Figure 1 are arbitrary. In the other hand, in Figures 2 to 5 there is a strict correspondence between the selected colors.*

State somewhere in the text at the beginning of section 3 that the 'arbitrary' color scheme from Fig.2 will be the same in all plots.

*The clarification regarding the choice of colors was included in Section 3 and in the legends to figures 1*

*and 2.*

l. 175: *"a reasonable separation of objects is achieved".*

This judgment is based on what property of the plot?

Also I disagree with the statement, Fig. 3 rather looks like a continuous scatterplot.

**(M2)**

Fig. 3: → Please show the corresponding probability density estimates of this scatterplot, maybethere are relative density maxima corresponding to some clusters. But from the current plot one can't say.

*We agree that this phrase is inappropriate and it was reworded. As is well known k-means is an iterative data partitioning algorithm that assigns n observations to exactly one of the 6 clusters previously established in the hierarchical CA and defined by six centroids. That is, k=6 is chosen before starting the algorithm. This concept is included in the paragraph preceding Fig. 2.*

Figs. 2, 4, 5, and 6: I strongly suggest to number the clusters according to the distributions in Fig. 3,from left to right. In Fig. 2, please add the corresponding cluster numbers which are missing.

*The numbering of each cluster in figures 2 to 5 is now indicated below Fig. 1.*

Fig. 6 and corresponding text:

Discussion missing, please at least discuss the different clusters in relation to different high DTfrequency regions from e.g. **Wilhelmsen et al. (2020)**

*In the revised version, the discussion section includes additional references to previous work, in particular from Wilhelmsen et al. (2020).*

In my opinion, describing the clusters as symmetrically distributed relative to the equator haslittle meaning: they are related to the subtropical jets on both hemispheres, so sure this will look somewhat symmetric, but it's the relation to the jets that has interpretative value.

*This point was highlighted in the discussion following Fig. 5 (old Fig. 6).*

> Fig. 6 looks quite coarse, since it works with monthly timeseries, a refinement to 5x5 degree orbetter is possible from RO coverage, and would be very welcome.

*We agree with this comment, however after testing different possible resolutions for the cells we concluded that the optimum size for monthly time series was 10x10 degrees.*

l. 204-205: *"The interconnected nature of each sub-region is highlighted in the polar, sub-polar andequatorial regions by clusters 1 and 4."*

**(M2)**

I'm sorry to put it this way, but to me this sentence is quite euphemistic. What I see is themethod's weakness: equatorial and polar DT's have very little in common, yet the clusteringmethod is mixing them up.

*We agree with this point. This comment was also modified and reworded, following Fig. 5.*

Clusters 1 and 4 are next to each other in Fig. 3, and their spatial distribution in Fig. 6 has no distinct structure, it makes me doubt about the method's ability to separate some DT features – withthe used settings in this manuscript. See below for further suggestions on the method.

**General comment on section 3:**

I am not sure the parameters used for the clustering analysis are the most meaningful. With the method as is, all DT types as well as all modes of variability are blended together and really difficult to separate. I have a couple of suggestions that should help with the clustering and interpretation of results:

- *We believe that the revised version discusses in some detail the choice of the parameters used, always considering the present work as a first stage of a more in-depth analysis.*

- Separating NDT' by time-scales, e.g. into subseasonal, interannual, QBO-specific, ENSO-specific… would make any cluster regions found easier to interpret, and relatable to previous works. For example, a good test of the clustering method would be to compare its output regions tothe El Nino – La Nina differences shown in **Wilhelmsen et al. (2020)**, their Fig. 3c.

  Combine with different parameters for clustering: e.g. DT depth (difference in height between the two tropopauses) or its standard deviation could be tested instead of std(NDT') to see whether the clustering performance improves. DT depth properties should distinguish equatorial and mid-latitude / polar DT much better than DT frequency alone.

I think some sensitivity tests with other parameters and/or case-studies (e.g. ENSO-specific timescale) are necessary in order to reassure the audience that this method can give some useful andmeaningful output, that can be related and build upon previous DT studies. Especially the current Figure 6, in my opinion is rather far from being convincing on the method's potential, and it's donewith a single setting for clustering on, again, basic distribution parameters of frequency.

Once clustering gives something more robust, one may expect a lot more juice coming from the multivariate regression analyses.

*As mentioned above, we agree with the potential interest of these possible new research directions, in addition to those suggested in our new section 4, concerning clustering and modeling. However, we prefer to leave them for a future contribution.*

**Sect. 4+5#**

I feel these sections could be summarized into a couple of methods subsections. They are very heavy in the present manuscript, their extended version with all the minutiae can be moved to anappendix or supplement.

Same with the first half of section 6 actually.

*The general and specific structure of the revised version is very different from the original one.*

l. 320-345: I don't think it's necessary to explain what is the $R^2$, adjusted $R^2$ and F-statistic, these are standard things… please explain the most relevant details for model evaluation in a very summarized way.

*This was modified in Section 4.*

**Sect. 6#**

l. 379-385: I was thinking exactly this while reading everything after section 3: the clustering analysis feels to me like the bottleneck for the rest of the methods/results – it is imperative that theresults in section 3 are robust and thoroughly tested, then sections 4-6 will benefit greatly and interpretation beyond just the statistics will be more straightforward.

l. 387-403: this is the only paragraph where the results from the multivariate linear regression are discussed, this part should be markedly extended.

From these results, please highlight what is added upon the referenced work.

From the results of this section, can you say something about which climate indices affect STEthe most? Can your methodology provide a way forward to answer such question?

l. 430-435: I don't think it's necessary to explain what a q-q plot is…

*This was modified and reduced in Section 4.*

Figs. 10-11:

Both appear a bit pixelated, please increase quality, and reduce the horizontal separation between the red boxes. Also, titles are repeated in each sub-panel, you can save a lot of space using one title above all panels.

*Done.*

l. 439-445: this can be considered an outlook, please create a separate section 'Summary & Outlook' or similar that summarizes main results and discusses possible applications and adaptability of your method.

*The revised version, in particular section 4 was rewritten. This suggestion has been included.*

**Appendix C**: is it really necessary that these very technical infos stay within the main manuscript? I'd suggest to move it to a separate supplement document.

*We believe that the appendixes are not strictly contained in the main manuscript, but we have no*

*problem to create a separate supplement if necessary.*

**Data availability statement:** state here (or reference in methods section) what software is used forclustering analysis and regression model.

*Done.*